evolution, physiology, genetics

ketocarotenoid, pigmentation, ornamental coloration, androgens, gene expression

**Author for correspondence:**
Sarah Khalil
e-mail: skhalil@tulane.edu

# Testosterone regulates *CYP2J19*-linked carotenoid signal expression in male red-backed fairywrens (*Malurus melanocephalus*)

Sarah Khalil[1], Joseph F. Welklin[2,3], Kevin J. McGraw[4], Jordan Boersma[5], Hubert Schwabl[5], Michael S. Webster[2,3] and Jordan Karubian[1]

[1]Department of Ecology and Evolutionary Biology, Tulane University, New Orleans, LA, USA
[2]Macaulay Library, Cornell Laboratory of Ornithology, Ithaca, NY, USA
[3]Department of Neurobiology and Behavior, Cornell University, Ithaca, NY, USA
[4]School of Life Sciences, Arizona State University, Tempe, AZ, USA
[5]School of Biological Sciences, Washington State University, Pullman, WA, USA

SK, 0000-0002-6929-4540; JFW, 0000-0002-7534-8795; KJM, 0000-0001-5196-6620; JB, 0000-0001-8355-2027; HS, 0000-0001-8253-0133; MSW, 0000-0001-7585-4578; JK, 0000-0001-8201-9992

Carotenoid pigments produce most red, orange and yellow colours in vertebrates. This coloration can serve as an honest signal of quality that mediates social and mating interactions, but our understanding of the underlying mechanisms that control carotenoid signal production, including how different physiological pathways interact to shape and maintain these signals, remains incomplete. We investigated the role of testosterone in mediating gene expression associated with a red plumage sexual signal in red-backed fairywrens (*Malurus melanocephalus*). In this species, males within a single population can flexibly produce either red/black nuptial plumage or female-like brown plumage. Combining correlational analyses with a field-based testosterone implant experiment and quantitative polymerase chain reaction, we show that testosterone mediates expression of carotenoid-based plumage in part by regulating expression of *CYP2J19*, a ketolase gene associated with ketocarotenoid metabolism and pigmentation in birds. This is, to our knowledge, the first time that hormonal regulation of a specific genetic locus has been linked to carotenoid production in a natural context, revealing how endocrine mechanisms produce sexual signals that shape reproductive success.

## 1. Introduction

Carotenoid pigments, which provide many of the vivid red, orange and yellow colours observed in vertebrates, have long captured the attention of behavioural ecologists interested in the evolution of social signals. Among vertebrates, carotenoid-based colours can serve as honest indicators of quality because of costs associated with obtaining and producing them, inspiring classic research surrounding the adaptive benefits of sexual signalling [1,2]. However, many aspects of the underlying mechanisms of carotenoid colour production remain unclear [3] (though see [4,5]), even though many assumptions surrounding honest signalling rely on understanding these proximate mechanisms. For example, carotenoids can be important for both antioxidant and immune functions [6,7], which suggests a trade-off between using carotenoids for physiological maintenance versus signal production [8,9]. However, recent work has cast doubt on the degree to which carotenoids function in physiological defences [10] or trade-off with somatic maintenance [11], sparking debate about the costs associated with production or maintenance of

carotenoid-based signals [12,13]. Resolving the underlying proximate mechanisms involved in carotenoid production can help inform this debate by improving our ability to interpret how multiple physiological pathways interact to shape and maintain carotenoid signals [3].

Once acquired from the diet, carotenoids can be modified via endogenous enzymatic processes that modulate carotenoid-based signal expression [14–17]. In contrast to some endogenously synthesized pigments such as eumelanin [18,19], researchers have only recently begun describing and characterizing the enzymes, genes and pathways involved in carotenoid metabolism and deposition [15,20,21]. Carotenoid ketolation, which includes the metabolism of yellow dietary carotenoids into red ketocarotenoids [22], is an important innovation in vertebrate evolution and colour diversification [23–27]. In birds in particular, many well-known examples of sexually selected visual displays involve bright-red keto-carotenoid-based coloration in the plumage or bare parts (e.g. [28–31]). Two recent studies have independently identified and described the locus *CYP2J19* as a gene that codes for the production of a putative ketolase enzyme (from the cytochrome P450 family) that underlies ketocarotenoid pigmentation in birds [14,15]. Though the discovery of this ketolase has important implications for understanding carotenoid signalling, the generality of this mechanism remains unclear since these studies have so far linked *CYP2J19* to red coloration in aberrantly coloured domesticated birds: the mutant 'yellowbeak' zebra finch (*Taeniopygia guttata*) and a hybrid breeding line (red-factor) of canary (*Serinus canaria*). However, recent work has suggested that *CYP2J19* expression may account for interspecific variation in red coloration—for example, *CYP2J19* expression was consistently higher in the liver of species of weaverbirds (Ploceidae) with red plumage compared to species with yellow plumage [24]. However, to date to our knowledge, no empirical study has linked *CYP2J19* expression to intraspecific variation in expression of red coloration in a wild species, limiting our ability to evaluate potential adaptive consequences of this gene-regulatory mechanism to sexually selected traits [32].

Endocrine regulation often underlies the development and expression of sexually selected traits [33]. Androgens, such as testosterone, play a fundamental role in mediating gene expression and resulting phenotype [34,35]. In many animals, testosterone is involved in phenotypic integration at the level of the individual, including determining breeding phenotype [34,35]. For example, in species characterized by complex social hierarchies, such as cooperative breeders [36,37], dominant breeding males often have elevated testosterone relative to subordinate 'helper' males [38–41]. Testosterone regulates changes in physiology and phenotype to help match organisms to their social environment, including changes in social rank and breeding status [42,43]. At the same time, studies in other systems provide little support for the phenotypic-integration role of steroid hormones [44,45], revealing limitations in our understanding of these relationships. Indeed, many key components of the endocrine-genomic mechanisms involved in regulating signal expression remain unresolved [46,47], especially for carotenoid-based signalling systems [48–50], highlighting the need for additional research into how phenotypic integration operates on a mechanistic level.

The red-backed fairywren (*Malurus melanocephalus*) provides a useful study system for linking these two lines of inquiry and assessing how *CYP2J19* and testosterone regulation may interact to control intraspecific variation in the expression of a carotenoid-based signal. Within a population of this cooperatively breeding species of bird, males can express either ornamental black body plumage with a carotenoid-based red dorsal feather patch, which is displayed in courtship, or female-like unornamented brown plumage [51,52]. Individual males can flexibly express unornamented or ornamented plumage depending on several factors including (i) breeding status: nearly all males moult into ornamented plumage during the non-breeding season (pre-alternate moult) in preparation for breeding, though timing and duration of this phase vary, and most males moult back into unornamented plumage after the breeding season (pre-basic moult); (ii) age: approximately 15% of males in our study population moult into ornamented plumage in their first breeding season, 90% by 2 years of age, and nearly all males by their third year; and (iii) physiological condition: males in better condition are more likely to moult into ornamented plumage [53]. Ornamented males have higher reproductive success than unornamented males, driven by higher rates of extrapair paternity in ornamented males [54]. In addition, there is experimental evidence for strong female preference for males with redder dorsal plumage [55], suggesting that this is a sexually selected trait in this species, which in turn appears to be driving introgression of redder plumage between subspecies [56]. Previous work also indicates that androgens (in particular testosterone) are important in signal acquisition: ornamented male red-backed fairywrens have higher levels of circulating androgens than unornamented males [57], unornamented males experimentally implanted with testosterone moult into the ornamental red/black plumage [58], and younger non-breeding helper males rapidly increase androgen levels and develop carotenoid-based plumage when experimentally provided with breeding opportunities [59]. In addition, testosterone treatment of females can induce some red plumage coloration in normally brown females that would otherwise have low circulating androgens [60]. Revealing how these endocrine processes may regulate gene expression associated with sexually selected carotenoid coloration has the potential to advance our understanding of the evolutionary origins and trajectories of carotenoid ornaments.

In this study, we test the hypothesis that testosterone regulates carotenoid-based plumage expression at the level of gene expression in male red-backed fairywrens. Specifically, we investigate the relationship between testosterone and carotenoid-based signal production by assessing the correlation between circulating ketocarotenoid concentration and plumage phenotype (i.e. expression of the carotenoid-based plumage patch), as well as experimentally test for the role of testosterone in regulating *CYP2J19* expression. We first use high-performance liquid chromatography (HPLC) to determine circulating carotenoid levels of ornamented red/black males, unornamented brown males, and females to identify the degree to which variation in metabolized ketocarotenoid circulation explains differences in carotenoid-based signal expression among these phenotypes. To evaluate the genomic regulation of these ketocarotenoid differences, we measure relative expression of *CYP2J19* in the liver (a central site for carotenoid metabolism [14,15,61,62]) in each phenotype. Finally, we experimentally test for the role of testosterone in regulating *CYP2J19* expression by measuring expression levels in testosterone-implanted unornamented males that developed carotenoid-based plumage in response to testosterone treatment. Our results are consistent with the hypothesis that, in red-backed fairywrens, testosterone mediates

expression of carotenoid-based plumage by regulating expression of *CYP2J19*.

## 2. Material and methods

### (a) Plasma sample collection and quantifying circulating carotenoids

We collected samples and conducted experiments (below) on free-living red-backed fairywrens captured in mist nets at our long-term study site in Samsonvale, Queensland, Australia (27°27′ S, 152°85′ E). We collected blood samples (20–70 µl) from the wing vein using heparinized microcapillary tubes from May to August 2017 and 2018 during the non-breeding season, a period when most males were actively moulting into their breeding season plumage. Blood was centrifuged for 5 min at 10 000 r.p.m., after which the plasma was separated from the packed cells and stored at −20°C until transport to the USA where samples were stored at −80°C until HPLC analysis.

Plumage score was recorded at time of capture following prior methods [63], yielding total ornamentation scores that ranged from 0 (brown) to 100 (red/black). Based on this score, males were labelled as either 'unornamented' (brown plumage, score < 33), 'intermediate' (mixed plumage, score between 33 and 66), or 'ornamented' (red/black plumage, score > 66). Females always have completely brown plumage (plumage score = 0) and are therefore considered unornamented. Timing of moult into ornamented plumage is variable (J. F. Welklin, S. M. Lantz, S. Khalil, J. Karubian, M. S. Webster 2020, unpublished data), similar to other recorded *Malurus* species [64], meaning that male plumage at time of capture and sample collection may differ from the 'final' plumage colour score the male expressed later in the breeding season. Males can breed in unornamented or ornamented plumage or serve as auxiliaries with unornamented plumage (helpers) at the nest [53]. Because we were interested in differences between unornamented and ornamented plumage, we documented the 'final' plumage score of colour-banded individuals on 1 November (the approximate mid-point of the breeding season) and used this 'final' score for analyses of unornamented males, ornamented males, and females; we excluded from analysis the relatively small subset of birds whose 'final' plumage score was intermediate ($n = 11$). We assigned either minimum or known age (age range 1–7 years, 77%, 123 of 160, were of known age) to all birds at the time of sample collection using nestling banding records or extent of skull ossification (ossification scale modified from [65], and we have validated this scale within this species multiple times). Qualitatively similar results were obtained in analyses run with these age criteria, or using only known-age birds (see the electronic supplementary material, table S1).

We used HPLC to identify and quantify the concentration of carotenoids in the plasma, following the methods of Rowe & McGraw [51]. We analysed carotenoids in 160 plasma samples ($n = 42$ females, $n = 29$ unornamented males, $n = 89$ ornamented males). To assess the relationship between circulating ketocarotenoid levels and plumage phenotype, we ran a linear mixed-effect model with the lme function in the R package nlme [66], R v. 3.6.0 [67]. The model included total circulating ketocarotenoid concentration (i.e. the sum of alpha-doradexanthin, astaxanthin, adonirubin and canthaxanthin concentrations) as the response variable and the following predictor variables: (i) phenotype (female versus unornamented male versus ornamented male); (ii) age (as a continuous variable); (iii) year of sample collection; and (iv) the interaction between phenotype and age. To control for repeated measures of the same individual across years, we added individual as a random effect ($n = 13$ individuals sampled both years). Residuals were inspected visually for homoscedasticity, and we used the varIdent function to control for heterogeneity of variance between groups. Year of sample collection did not improve model fit (i.e. it did not improve Akaike information criteria by more than 2 and the *p*-value of the year variable was greater than 0.05) and was therefore dropped. We tested the model for significance of phenotype with a Tukey's posthoc test using the glht function in the R package multcomp [68].

### (b) Testosterone implantation and liver sample collection

We collected liver samples from breeding, but not auxiliary helper, red-backed fairywrens in November 2017, to control for potentially confounding underlying differences in endocrine or genetic profiles that may exist between auxiliary non-breeding versus breeding individuals [57]. First, three breeding unornamented males were implanted with testosterone. At time of initial capture and implantation, around 10 feathers were plucked from the centre of the back to induce feather replacement at that location. Implants were composed of beeswax (73% by weight; Sigma-Aldrich, St. Louis, MO, USA) and hardened frozen peanut oil (24% by weight; ACROS Organics, NJ, USA) that were mixed in a water bath at 67°C. Once the beeswax/peanut oil mixture was melted, crystalline testosterone (3% by weight; Sigma-Aldrich, St. Louis, MO, USA) was dissolved in 2.5 µl of 200 proof ethanol (Fisher Bioreagants™), and the testosterone suspension was then added to the wax mixture and stirred. The implants were formed by feeding partially solidified wax through the tip of a syringe that was cut so the diameter was 2 mm, resulting in implants of 2 × 3.2 mm weighing between 19.8 and 20.7 mg. Testosterone concentration in the beeswax carrier was scaled to produce high physiological concentrations found in circulation during the breeding season [57]. Implants were inserted subcutaneously using forceps above the thigh into a small (2–3 mm) skin incision that was sealed with veterinary skin adhesive. After confirming that the incision was completely sealed and the bird was in good condition, the bird was released. Three unornamented males were implanted with sham controls (beeswax/peanut oil implant with no testosterone) and also had around 10 back feathers plucked.

Implanted birds were recaptured 10–12 days post-implantation for liver sample collection, a time period that allowed for growth of pin feathers in the plucked plumage patches (red pins in testosterone-implanted males, and a mix of red and brown pins in the sham-implanted males, consistent with what had been observed in another feather-plucking study in this species [59]). We were unable to recapture one of the sham-implanted birds after implantation. Instead, we captured and obtained samples from one additional breeding unornamented male, who was not implanted, to include in our control group. Following recapture on territories in mist nets, birds were immediately sacrificed by cervical dislocation. Body dissection was performed in the field, and the right lower lobe of the liver was removed and stored in 1 ml of RNAlater storage buffer (ThermoFisher Scientific), and immediately placed on dry ice. Samples were stored at −80°C until RNA extraction. In addition to the three testosterone-implanted unornamented males and the three control unornamented males (two with sham implants, one without an implant), we also collected liver samples for three ornamented breeding males and three breeding females (without implants). All birds sacrificed were seen paired with a male or female within two weeks prior to sample collection, and all samples were collected within a period of 9 days. Circulating androgens were measured using an established radioimmunoassay protocol for this species (full methods in [57,69]); the intra-assay coefficient of variation was 8.68%. Testosterone-implanted birds were confirmed to have high concentrations of circulating androgens at time of collection (mean = 3027 pg ml$^{-1}$, range = 2198–4065 pg ml$^{-1}$), which is within the natural range of androgens for breeding ornamented males in this species [57]. Ornamented males also had similarly high levels of androgens at time of

collection (mean = 1834 pg ml$^{-1}$, range = 1124–2925 pg ml$^{-1}$). We were unable to obtain samples to assay testosterone concentrations for control unornamented males or females.

### (c) Quantifying relative expression of CYP2J19

To extract messenger RNA (mRNA), we removed liver tissue from the RNAlater buffer and homogenized it in a Qiagen TissueRuptor. We used a Qiagen RNAeasy mini-kit, following manufacturer's instructions, and reverse transcribed the mRNA to complementary DNA (cDNA) with a Superscript IV first strand synthesis kit (Invitrogen). All quantitative polymerase chain reaction (qPCR) reactions were run on CFX96 Touch™ Real-Time PCR Detection System (BioRad) with CFX Maestro Software (BioRad), using PowerUp SYBR Green Master Mix (Thermofisher Scientific). For measurements of CYP2J19 expression, we used qPCR primers CYP2J2-2F and CYP2J2-2R [14]. We assayed gene expression in triplicate for each sample (except for one female sample where CYP2J19 was assayed in duplicate) and normalized the data using the housekeeping gene GAPDH, using primers Gg_GAPDH_qPCR_F and Gg_GAPDH_qPCR_R [15]. Reaction conditions for qPCR were tested and optimized using a standard curve produced by creating a serial dilution of a pool of all cDNA samples. Efficiencies ranged from 95% to 105%, and we analysed qPCR data using the delta-delta Ct method [70], further described in the electronic supplementary material, Methods.

We found no effect of the presence or the absence of the sham implant on gene expression within unornamented males (see the electronic supplementary material, Methods), and no evidence of homoscedasticity (Breusch Pagan test, $p > 0.05$). We tested for statistical differences in liver CYP2J19 expression (log fold change) between phenotypes with an ANOVA, using the aov function in R, followed by a Tukey's posthoc test using the Tukey HSD function in R.

## 3. Results

### (a) Circulating carotenoid concentration was associated with sex and plumage phenotype

We identified six different circulating carotenoids in fairywrens: two dietary xanthophylls, lutein and zeaxanthin, and four red metabolized ketocarotenoids previously identified in the red dorsal feathers of red-backed fairywrens [51]: alpha-doradexanthin, astaxanthin, adonirubin and canthaxanthin. Plumage phenotype was a significant predictor of total circulating ketocarotenoid concentration ($F_{2,48} = 240.2$, $p < 0.0001$; figure 1), where ornamented (red/black) males had higher concentrations of circulating ketocarotenoids than either unornamented (brown) males (Tukey's posthoc, $p$.adj < 0.001) or females (brown) ($p$.adj < 0.001). There was also a significant effect of the phenotype × age interaction ($F_{2,48} = 24.7$, $p < 0.0001$) but no effect of age alone (electronic supplementary material, table S2). Specifically, ages differed in ketocarotenoid circulation only for unornamented males, where 2-year-old unornamented males had higher concentrations of circulating ketocarotenoids than their 1-year-old counterparts (figure 1 and the electronic supplementary material, figure S1).

### (b) CYP2J19 expression in the liver was associated with expression of red plumage

Liver expression of CYP2J19 differed significantly among plumage phenotypes ($F_{3,8} = 114$, $p < 0.0001$; figure 2). Ornamented males had higher relative expression of CYP2J19 than did unornamented males (Tukey's posthoc, $p$.adj = 0.038), and unornamented males had higher relative expression of CYP2J19 than did females ($p$.adj < 0.0001).

### (c) Testosterone upregulated CYP2J19 expression

Testosterone-implanted unornamented males had significantly higher expression of CYP2J19 in liver tissue than did control unornamented males (Tukey's posthoc, $p$.adj = 0.012; figure 2) or females (Tukey's posthoc, $p$.adj < 0.0001), with no difference in expression between testosterone-implanted unornamented males and ornamented males (Tukey's posthoc, $p$.adj = 0.83).

## 4. Discussion

By combining field-based observational and experimental investigations with gene expression and biochemical analyses, we found that testosterone regulates gene expression implicated in the production of sexually selected red plumage in male red-backed fairywrens. Ornamented males with carotenoid-based red plumage ornamentation had higher levels of circulating ketocarotenoids than either females or unornamented males (both with brown plumage). Carotenoid metabolism is an endogenous process, and the ketolase enzyme encoded by CYP2J19 has previously been demonstrated to convert dietary carotenoids into red ketocarotenoids in birds [14,15]. Along with unmanipulated ornamented fairywren males having higher expression of CYP2J19 in the liver than either unornamented males or females, our experiment confirmed that elevated testosterone levels resulted in increased CYP2J19 expression in unornamented males. These findings are supported by field studies showing that ornamented plumage is testosterone-dependent in red-backed fairywrens [57,58]. Taken together, these results are consistent with a mechanistic hypothesis (figure 3), whereby circulating testosterone levels change in males in response to intrinsic or extrinsic cues (breeding status, age, physiological condition [53,57]), which in turn modifies expression of CYP2J19 in the liver and thus increases concentration of metabolized ketocarotenoids in the plasma, which is associated with red plumage ornamentation. Though we did not test this full mechanistic hypothesis within the same individuals (i.e. it is based on two separate datasets, one for circulating ketocarotenoids and one for the testosterone-gene expression experiment), this proximate pathway for carotenoid-based signal production establishes a link between the endocrine and gene-regulatory system and avian coloration important for sexual signalling and elevated male reproductive success.

Testosterone has been shown in several studies to be an important activator of male ornamentation [33,71,72], yet there is little work investigating how testosterone and ornament production are linked [73,74]. Mutations and expression differences in CYP2J19 have been implicated in driving carotenoid trait divergence between species and subspecies [14,15,23,24,75,76], but it has not previously been shown whether CYP2J19 expression levels influences intra-population variation in ornamentation. Our study design focused on characterizing differences in plumage phenotypes within a wild population, revealing how hormonal regulation of CYP2J19 expression generates variation in a polymorphic sexual signal. Past studies combining endocrine and gene expression analysis in other natural systems have revealed how flexibility in behaviour is mediated within a species

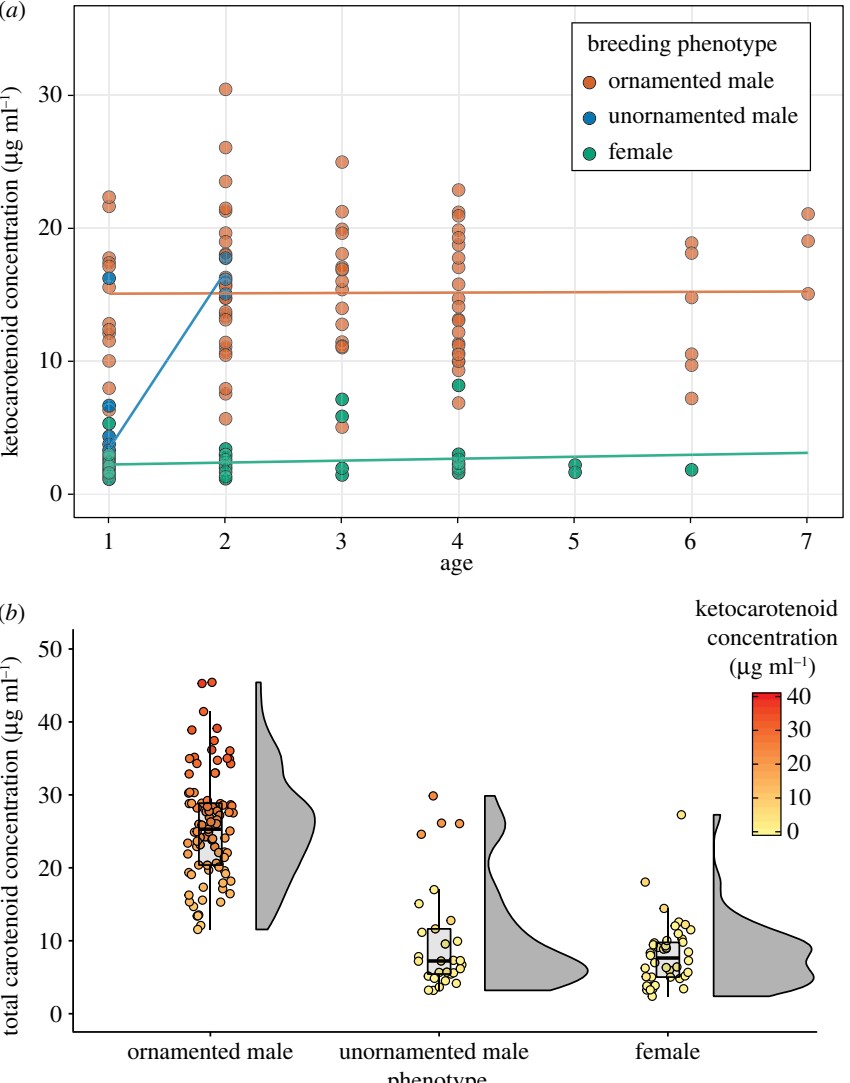

**Figure 1.** Ornamented male red-backed fairywrens have higher concentrations of circulating metabolized ketocarotenoids than do either unornamented males or females. (*a*) Scatterplot points show ketocarotenoid concentration as a function of breeding phenotype and age. Lines represent fitted lines from the linear mixed-effect model described in text. (*b*) Total plasma carotenoid concentration for the different plumage phenotypes are presented as boxplots, indicating the median and quartiles with whiskers reaching up to 1.5 times the interquartile range. The half-violin plot outlines show kernel probability density. The scatterplot points show the value of the total carotenoid concentration for each sample, and the colour of the point represents the concentration of metabolized ketocarotenoids in that sample, represented as a scaled gradient where yellow points have low metabolized ketocarotenoid concentration and red points have high metabolized ketocarotenoid concentration (see inset legend). (Online version in colour.)

[77,78], and how phenotypic differences may evolve between species [79], improving our understanding of hormone-mediated phenotypic evolution. The current study adds to this body of work by suggesting that incorporating endocrine control of gene expression to studies of colour production may be a rewarding research avenue for assessing signal evolution. Specifically, our findings show support for a role of testosterone as a transcriptional regulator of sexually selected phenotypes, which may also be important in other species with strong sexual selection for ornamented phenotypes.

Testosterone may also help explain two unexpected results from this study. First, why would unornamented males have higher expression of *CYP2J19* than females, despite both of them having similar brown plumage? Unornamented male red-backed fairywrens that shift from being an auxiliary helper to being a breeder during the breeding season exhibit rapid increases in their circulating testosterone levels [59], and in general male breeders have higher levels of circulating testosterone than do helpers [57]. These higher concentrations of testosterone probably benefit unornamented breeders by

mediating observed increases in breeding-specific behaviours such as territory defence [63,80], but also probably trigger increased *CYP2J19* expression. Whether this increase in *CYP2J19* expression is merely a by-product of testosterone or whether it somehow benefits these males is unclear, but one possibility is that it could allow newly paired unornamented males to quickly start depositing red ketocarotenoids into their developing feathers if they are still in the moulting window. Second, why do we find an age effect on circulating ketocarotenoids among unornamented males, where 2-year-olds have higher concentration than 1-year-olds? Here again, testosterone may mediate differences in *CYP2J19* expression that underlie these results. Specifically, 1-year-olds are more likely to serve as helpers in their natal group and hence have lower levels of circulating testosterone, whereas most 2-year-olds transition into a breeding role, even after having completed moult into unornamented plumage [63,81], which may explain this shift in carotenoid concentration between the two age groups. Future work with greater sampling will reveal how tightly linked age, testosterone and gene

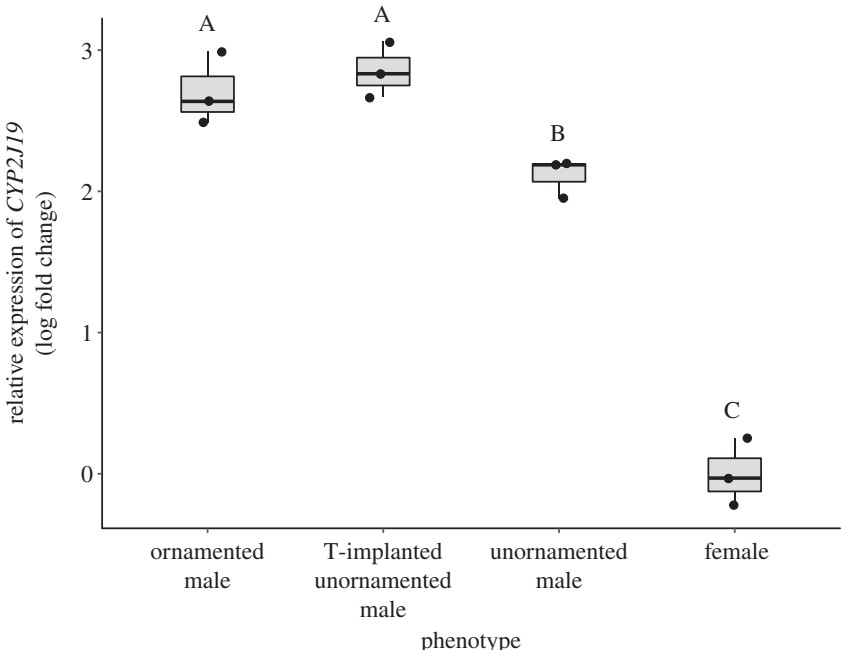

**Figure 2.** Testosterone implantation upregulates the expression of the *CYP2J19* gene in the liver of unornamented males. Shown are qPCR measurements of expression of *CYP2J19* relative to housekeeping gene *GAPDH* in ornamented males, testosterone (T)-implanted unornamented males, control unornamented males, and females of the red-backed fairywren. Points represent samples from individual birds, and boxplots indicate the median and quartiles with whiskers reaching up to 1.5 times the interquartile range. Different letters above plots indicate significant differences at $p < 0.05$ by Tukey's HSD.

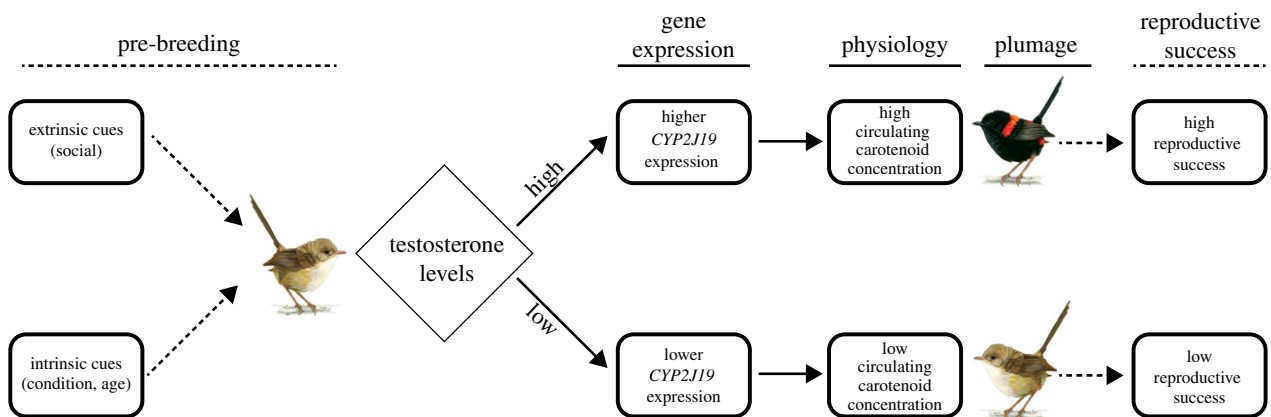

**Figure 3.** A working mechanistic model for the acquisition of carotenoid-based plumage in male red-backed fairywrens (modified from [53]). Under this model, extrinsic and intrinsic cues interact to affect testosterone levels. In turn, testosterone levels mediate expression of the *CYP2J19* gene (and probably other genes), which appears to elevate circulating ketocarotenoid levels in plasma, which in turn are linked to plumage phenotype. Solid arrows depict connections supported by this study, whereas dashed arrows depict connections supported by previous work in this system. Illustrations of fairywrens by Allison Johnson. (Online version in colour.)

expression are to each other, and how often ketocarotenoids are produced even when not deposited into the plumage.

Our results corroborate a growing number of studies suggesting the importance of *CYP2J19* for red coloration in birds [23,24,82,83] and other animals [84,85]. The genetic basis of carotenoid-based coloration is not yet well resolved [3,32], but a handful of genes with large effects have been identified in recent years that are associated with carotenoid coloration (e.g. *BCO2* [86–88], *SCARB1* [16], *CYP2J19* [14,15] and other cytochrome P450's in other taxa [89,90]). There are probably many more genes involved in processes such as carotenoid uptake, processing and transport [91], all of which might influence coloration to varying degrees. Yet, without a clearer understanding of how these genes and other underlying processes interact to produce carotenoid phenotypes, our ability to identify and interpret the selective forces and

evolutionary processes that maintain these phenotypes remains limited. One promising route to identifying genes associated with carotenoids is to use RNA sequencing, especially in the context of experimental manipulation of phenotypes as done here with hormones, to identify gene pathways that underlie red coloration. We note the small sample size for our gene expression experiment ($n = 3$ per phenotype, owing to permitting restrictions), and we attempted to minimize any effect of this by only using breeding individuals to reduce phenology differences as well as collecting all birds in a relatively small time period (described in methods). Additionally, future work might test for the role of *CYP2J19* activity in other tissues relevant for colour signal production, including feather follicles and other integumentary tissue (e.g. bill or leg keratinocytes) [92], and investigate finer-grained relationships between *CYP2J19* expression and hue of red

plumage to extend beyond the presence/absence-of-coloration approach we employ in the current study. An integrative approach to studying colour signals by combining new research on proximate mechanisms of signal production with our current understanding of ultimate explanations for these signals (e.g. importance in mate choice and reproductive success), will improve our broader understanding of how such traits are regulated and shaped by selection.

Ethics. All procedures in this study were approved by the Tulane University Institutional Animal Care and Use Committee (IACUC 2019-1715), Cornell University IACUC (2009-0105), Washington State University IACUC (ASAF no. 04573), the James Cook University Animal Ethics Committee (A2100) and under a Queensland Government Department of Environment and Heritage Protection Scientific Purposes Permit (WISP15212314).

Data accessibility. Data available from the Dryad Digital Repository: https://dx.doi.org/10.5061/dryad.pnvx0k6jp [93].

Authors' contributions. S.K. contributed to conceptualization, writing, editing, data collection and funding acquisition, and carried out qPCR laboratory work, formal statistical analysis and visualization. J.F.W. contributed to data collection and editing. K.J.M. carried out HPLC laboratory work and contributed to editing. J.B. made testosterone implants, carried out testosterone assays and contributed to editing. H.S. contributed to conceptualization, data collection and editing. M.S.W. contributed to conceptualization and editing. J.K. contributed to conceptualization, supervision, editing and funding acquisition.

Competing interests. We declare we have no competing interests.

Funding. This work was supported by the National Science Foundation (IOS-1354133 and IRES-1460048 to J.K.) and Tulane University Department of Ecology and Evolution (to S.K.). S.K. was supported by an NSF Graduate Research Fellowship during part of this work.

Acknowledgements. We gratefully acknowledge field technicians Mary Margaret Ferraro, Maria Smith, David Weber, Sarah Duff and Malcom Moniz for their assistance in collecting samples in the field, as well as William Feeney and Matthew Marsh for assistance in the field. We also thank Southeast Queensland Water for access to land where we performed our field research. This manuscript was greatly improved by discussion and feedback from E.D. Enbody, N.R. Hofmeister, J. Walsh, A.C. Demery, I.J. Lovette, A.R. Gunderson and K.G. Ferris, as well as the Karubian laboratory. Illustrations of fairywrens by Allison Johnson.

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
