## [Reviewer comments · Proceedings of the Royal Society B: Biological Sciences]

Review History

RSPB-2019-2630.R0 (Original submission)

Review form: Reviewer 1

Recommendation

Accept with minor revision (please list in comments)

Scientific importance: Is the manuscript an original and important contribution to its field?

Excellent

General interest: Is the paper of sufficient general interest?

Good

Quality of the paper: Is the overall quality of the paper suitable?

Excellent

Is the length of the paper justified?

Yes

Should the paper be seen by a specialist statistical reviewer?

No

Do you have any concerns about statistical analyses in this paper? If so, please specify them explicitly in your report.

No

It is a condition of publication that authors make their supporting data, code and materials available - either as supplementary material or hosted in an external repository. Please rate, if applicable, the supporting data on the following criteria.

Is it accessible?

Yes

Is it clear?

Yes

Is it adequate?

Yes

Do you have any ethical concerns with this paper?

No

Comments to the Author

Here, Khalil et al. describe an empirical study that demonstrates the role of testosterone in mediating the expression of the “redness” gene, which in turn influences red ornamental colouration in male fairywrens. The greatest strength of this study is that the results don’t settle for correlations between natural levels of variation, but also suggest a direct causal relationship (through experimentally manipulation of testosterone in a subset of birds). The red-backed fairywren is a particularly amenable system for such questions because the testosterone-colouration relationship is perhaps clearer than in other birds with carotenoid-based colouration (where studies, I think, get a bit muddier and contradictory). Overall, I found this to be a clear, convincing, and fascinating manuscript, and my comments are mostly just comments, with a few points for extra clarification.

First paragraph: I found this to be one of the most clear, succinct, and accurate summaries of the carotenoid literature I’ve read in a recent manuscript. (And actually, the entire introduction is remarkably readable and informative!)

Lines 96-98: When I first learned about this system, I thought that any male in any given year could be either brown or ornamented morph – which, I suppose, is more or less true. But, I remember being surprised to learn that brown males are essentially always younger males and that ornamented males rarely (?) return to the unornamented breeding morph. If any of this is accurate, I think it’s worth clarifying briefly in the text because it does change how we interpret colouration, signalling, and polymorphism in this species.

Lines 166-167: Maybe this is described in results, but how many birds were measured in multiple years?

Line 176: Does it really count as a “moult” if the birds are just growing back that small region of feathers?

Line 188: I imagine there’s a story here... what happened to those birds? Or were they just too crafty to recapture?

Line 197: But the testosterone-implanted males would have been growing in an ornamented patch of feathers, right?

Line 202: How does this compare to unimplanted (but ornamented) males?

Lines 225-233: Just to clarify: you didn't take plasma carotenoid measurements from any of the implanted or sacrificed birds, right? So you didn't also test the effect of implantation on carotenoids? Or (and see next comment) are the implanted birds grouped with the ornamented males? [This comes up again on 262-264, where it's not entirely clear if the full pathway of T -> CYP2J19 expression -> ketocarotenoid concentration -> colouration was measured in the implanted, sacrificed birds, or whether different pieces of the puzzle were put together from different birds (which is, of course, quite fine)]

Lines 236-239: The figure clarifies this, but I think it's worth also clarifying in this text where the implanted birds are categorized (as ornamented males?).

Discussion: It could be interesting (and relevant) to mention that Koch, Josefson, Hill 2017 (Biol Rev) propose mito-carotenoid-hormone mechanistic connections, and Hill et al. 2019 (RSPB) provides some empirical evidence for the mito-carotenoid-CYP2J19 connection... Now, your study of course doesn't deal with mitochondria, but it does begin to flesh out the hormone-carotenoid-CYP2J19 connection – which in turn provides some insight into the mito relationships as well!

Review form: Reviewer 2

Recommendation

Reject – article is scientifically unsound

Scientific importance: Is the manuscript an original and important contribution to its field?

Acceptable

General interest: Is the paper of sufficient general interest?

Acceptable

Quality of the paper: Is the overall quality of the paper suitable?

Poor

Is the length of the paper justified?

No

Should the paper be seen by a specialist statistical reviewer?

No

Do you have any concerns about statistical analyses in this paper? If so, please specify them explicitly in your report.

Yes

It is a condition of publication that authors make their supporting data, code and materials available - either as supplementary material or hosted in an external repository. Please rate, if applicable, the supporting data on the following criteria.

Is it accessible?

N/A

Is it clear?

N/A

Is it adequate?

N/A

Do you have any ethical concerns with this paper?

Yes

Comments to the Author

Comments to editors

In my view, this article addresses an interesting question, the relationship between ketolase gene expression level and red colouration intensity at the within-species level. The article, moreover, proposes to test the testosterone influence on this gene, which is a nice idea. Accordingly, the title suggests that authors have demonstrated that the expression of the ketolase gene involved in yellow-to-red carotenoid conversion is influenced by testosterone levels. However, this assertion is based in a poorly made experiment with very low sample size (only three implanted birds were studied and compared to non-manipulated birds). Any indirect effect (e.g. age, condition) or merely choice can have deeply affected the result. Other methodological problems are also described below.

Major points

1. The model species seems to be quite particular. The authors previously published a study (Lindsay et al 2011 PlosOne) where 2-year-old males were implanted with testosterone during the pre-breeding moult period (August-Sept). Eight T-males produced a redder plumage than seven same-age implanted controls. They argued that this constitutes a particular case where testosterone does not inhibit moult, such as it occurs in other bird species where only a post-breeding moult takes place. Surprisingly, in contrast to that study, the authors now implanted their birds in the middle of the breeding season (November). Why this?
2. Perhaps to cope with this problem (no active moulting period), they plucked 10 feathers from the back. This would mean that birds would be stimulated to produce red ketocarotenoids for feathers. However, it is not clear if all the bird were able to moult the plucked feathers. Authors only mentioned that the 10-12 d period between implanting date and recapture is a time period that "allowed to re-growth" plucked feathers. Please, clarify.
3. Importantly, since the three sham-operated controls (also plucked) were not recaptured, other non-manipulated birds were used as controls. However, here we cannot discriminate if the supposed increase in CYP2J19 expression among the three T-males was due to high testosterone levels or to the fact of being engaged in producing new feathers. In other words, why should we expect that any of those birds captured only once were activating the ketolase to produce any pigment to feathers if they were not moulting at all?

Moreover, we cannot be sure how the surgery could have influenced the synthesis of redder carotenoids (which are powerful antioxidants).

4. I was also unable to know the age of those few birds used in the implant experiment and subsequent comparisons. The precedent correlational study on 160 birds reported an important role for age. The S1 fig suggests that older birds were abler to circulate red ketocarotenoids (see also de legend). Then, the experimental birds should be same-age birds to be compared, reducing this source of variability.
 5. Were birds used as controls in the experiment chosen non-randomly? If exactly three birds were used for comparisons, this suggests that authors selected some birds from their larger captures. What was the criterion here? This is important as this could induce some unconscious bias.
 6. Recent studies are showing that CYP2J19 can also be expressed at the follicles, not only at the liver. Authors must review Hanlu Twyman's studies (including his public PhD Thesis; see table 6.6.) where different bird taxa were assessed. This means that CYP2J19 expression at the liver could not be so relevant. Surprisingly, authors had the opportunity of detecting and quantifying the gene in re-growth follicles, but no information is reported.
- The site of conversion is also relevant to interpret the results of the correlational study on plasma

carotenoid values. Were red ketocarotenoids in plasma the same molecules giving colour to feathers or could they be perhaps subsequently transformed at the follicles?

7. The higher circulating level of red ketocarotenoids in red (ornamented) birds (Fig S1) could be a consequence of phenology. Could the moulting period be established at the individual level (maybe by flight feathers)? In that case, phenotype differences in the days elapsed from capture to the start of the pre-breeding moult should be tested. For instance, dull males could have been sampled farther from their individual moulting time, leading to lower circulating carotenoids. In that case, blood carotenoid levels would not be related to feather colour but to differences in phenology.

8- The birds were captured in two consecutive years in the correlational study on circulating carotenoids. Since the year factor is significant (Table S1), the authors should report sample sizes for each phenotype class each year, providing some chi-q test to know if they were imbalanced. In such a hypothetical case, phenotype-related effects could be, at least partially, due to year variability. If true, year x phenotype should be tested.

9. The significant interaction between phenotype and age in the same model is very relevant as it affects those conclusions only taken from analyzing the phenotype factor. This must be addressed in the main text, with some figure. What is the change in the slope between plasma total red carotenoids and age among the phenotypes? and what it means? Fig S1 is not a suitable representation. Authors must report dispersion plots with slopes for each phenotype type separately, and in the main text (I assume that age was a covariate, i.e. not a 7-level class variable). In the present version, nothing can be concluded about this interaction. I can only suspect that a positive correlation exists between carotenoid levels and age, and the slope is steeper among dull birds, older animals being able to circulate more red carotenoids. That influence of age is relevant to understand the testosterone experiment (pointed above). Please, note that precedent Lindsay et al 2011 demonstrating the influence of exogenous testosterone on feather colouration was made on second-year birds only.

10. The statistical test performed to interpret the testosterone manipulation engaged 12 individuals and a 4-level factor. Degrees of freedom are however 3,7 (L237). Why this?

11. Moreover, parametric ANOVA requires to meet the normality assumption. Was this the case? With so reduced sample size (particularly in pairwise comparisons: $n = 3$ vs 3) perhaps non-parametric tests should be preferable. In any event, in my view, the sample size is so short that any indirect uncontrolled effect (above) could strongly have affected the results (or merely choice).

An ethical note could here be considered as liver gene expression requires animal killing. However, the accepted rules of bioethics emphasize the fact that sample sizes should be enough large to avoid obtaining inconsistent results requiring to repeating the experiment. Moreover, it is surprising that CYP expression in feather follicles was not explored, which should remove any ethical concern.

12. How feather carotenoid composition was related to plasma composition? are red ketocarotenoids in blood the same that those in feathers? This is important to deduct if other transformations are also made in follicles. In that case, CYP2J19 should also have been assessed there.

13. What are the units in Fig 2? authors mention that they test delta-delta cq, but was is this? the simpler deltacq should be represented. Is the Ct reversed to directly indicate the gene expression level?

Other points

1. L57-59. Please, rewrite. Unclear.

2. L81. Endocrine regulation "underlines" phenotypes.... Please, clarify the meaning.

3. In my view, Peters 2007 BioEssays review on the integration of carotenoid-based signalling and testosterone-controlling effects on sexual signals should be checked and perhaps cited (DOI 10.1002/bies.20563)
4. L142-145. Taking into account the apparent phenotype variability, was the scoring method repeatable? I nonetheless understand that authors removed from the analyses some (11) birds classified in the middle of two classes, which should avoid any doubt.
5. Please, provide a reference to briefly explain how skull ossification was used to know the age of the birds.
6. L156. Analyses containing only known-age birds would have lower stat power (lower sample size; less 23%). Therefore, they can provide quite different results. Please, report them at least as supplementary material.
7. L177. How the peanut oil was hardened? by low temperature?
8. L203 the authors only provide testosterone levels for T-implanted birds at recapture date. Were these birds blood sampled at the first capture? this would allow knowing if T-levels indeed increased at the individual level. Anyway, plasma T-levels of other male birds should also be reported to compare them (though see comments on low sample size and stats).
9. L225-226. In the correlational study on circulating carotenoid levels, the sample of ornamented versus unornamented males was biased to the former 3:1. How was this considered statistically? The authors should test homoscedasticity. Moreover, authors should test normality as carotenoid concentrations are often biased with few animals reporting very high levels. This could require log-transformations. Nothing is commented in this regard.
10. L232. $p < 0.1$ does not mean a significant effect. Is this a mistake?
11. The authors should report the full information about the model used to test differences in circulating carotenoid levels. A table including F and P values for each term in the model with df should be reported, including some stat for the random term (was it significant?).
12. Fig s2. in the X-axis it should be mentioned "T-implant unornamented male"
13. Fig. 3. If the condition could influence testosterone levels and hence CYP, why body condition (size-corrected body mass) was not tested in the model on 160 birds?
14. Moreover, why testosterone in plasma was not assessed in these birds?

Review form: Reviewer 3

Recommendation

Major revision is needed (please make suggestions in comments)

Scientific importance: Is the manuscript an original and important contribution to its field?

Good

General interest: Is the paper of sufficient general interest?

Excellent

Quality of the paper: Is the overall quality of the paper suitable?

Good

Is the length of the paper justified?

Yes

Should the paper be seen by a specialist statistical reviewer?

No

Do you have any concerns about statistical analyses in this paper? If so, please specify them explicitly in your report.

Yes

It is a condition of publication that authors make their supporting data, code and materials available - either as supplementary material or hosted in an external repository. Please rate, if applicable, the supporting data on the following criteria.

Is it accessible?

Yes

Is it clear?

Yes

Is it adequate?

Yes

Do you have any ethical concerns with this paper?

No

Comments to the Author

General comments

This is a very interesting and necessary study that uses experimental approaches to demonstrate the mechanistic link between previously observed effect of T on carotenoid color. The results show that testosterone affects CYP2J19 expression unornamented males, and that, because CYP2J19 is involved in carotenoid synthesis, and plumage color correlates with carotenoid synthesis, that this gene provides a link between T and plumage color. The story would be more complete and stronger, if we could link the level of CYP2J19 expression to the actual plumage score. This is a test I think that the authors can do - by putting the phenotypes into categories, potentially interesting variation is lost. With small sample size, such a correlation may be hard to detect, however, especially since most of the variation in CYP2J19 is explained by sex (see below an additional comment on this).

An additional area where authors could make this study stronger is by being more specific about the implications of this study. The phenotypic integration angle is great, but I think that often in the general literature it is used as a vague umbrella term. I would ask the authors to speculate exactly how defining specific genes that regulate traits may contribute to the understanding of phenotypic integration and the evolution of integrated traits.

Specific comments

Abstract:

Great abstract, very succinct and clear. The only thing I am not sure about what are the “new avenues of investigation” for the link between hormones, signaling, and reproductive success. My understanding is that the link between T and color was shown before - what new avenues are you proposing given that now you have identified one of the candidate genes that may mediate this effect? Studies on the evolution of this gene or its promoters and enhancers? This may be addressed in the discussion, but it would be nice to mention that here, so that the last sentence is not a just-so statement.

Introduction:

50 - I would rephrase “less attention...” to something in the line of “more could be done” - I think that there has been plenty of research on the mechanisms of carotenoid production, e.g. Badyaev’s papers (which hasn’t been cited here and probably should).

62 - I would argue that we still don’t really know how certain melanin types are made (e.g. pheomelanin). You can solve this by saying “some endogenously synthesized pigments such as eumelanin” and then cite a study.

76 - CYP2J19 expression in liver? Or developing feathers?

79 - I agree that studies of wild birds are important, but I would still like you to explicitly say why.

90 - Add that understanding the mechanisms for carotenoid color regulation may provide better

insights into how phenotypic integration works. Also, you are painting a very rosy picture here about integration, but here have been some studies that have found that some hormones show poor evidence of integration (e.g. Garamszegi et al 2012 Ethology). Which makes it even MORE important to study the mechanisms.

93-96 – Move citations to the end of the sentence?

99 – maybe rephrase to “driven by the higher rates of extrapair paternity in ornamented males”

109-110 – I think a stronger way to say this would be “...these endocrine processes regulate gene expression”, as steroid receptors are transcription factors.

111 – Insert commas around “known a priori....selection”. but that may be a personal preference – it may make the sentence a bit less jammed.

L113 – change “explore” to “test”

113-128 – is there any way you can get data about the effect of T on the actual ketocarotenoid concentration? Or the correlation between T and ketocarotenoids? That would unite the two pieces of information here – you have correlation between ketocarotenoids and plumage, and the effect of T on CYP2J19, but it would be great if you can relate T to ketocarotenoids. If yes, can you cite it here? This is not a make it or break it, but if you have the data it would be great to see them.

Methods

152 – were the intermediate males that you excluded still molting? Not clear here. If they are done molting, why did you exclude them from the analysis?

167 – why “both”? It reads like you are running just one model. What is the difference between the two?

184 – does reference 49 validate these implants in this species? I am not sure how you knew that the implants you gave the birds resulted in the increase of T close to the breeding-season levels. But I appreciate the fact that you are trying to keep the hormone levels in reasonable biological range.

184 – did you use trocar syringe or forceps to insert the implant?

203 – how do these values relate to the natural concentration of T in the breeding season?

214 – did you validate that GADPH expression did not differ between treatments?

215 – is CYP2J19 the only GOI you tested using qPCR?

Results

235 – looking at the figures, it seems that there is a general effect of sex on the CYP levels.

However, that is not something you can explicitly test using your current anova model design, where the males are separated into 3 groups. You have a small sample size, so it may decrease your power, but I wonder what results you’d get if you ran a lm with sex being separate from the plumage score? It would get tricky because you only have T implants in the unornamented males, so the model might have to be nested with respect to that. I understand why you are doing anova here as you do, but each group in your current anova is identified by 3 factors (sex, plumage, and T treatment). The tukey approach allows you the test the T treatment of course, but it doesn’t allow an overall sex comparison. But I also understand that the sex comparison here is not the primary objective.

Discussion

General comment: I would like you to discuss the fact that the CYP2J19 levels are higher in the unimplanted unornamented males compared to the females – i.e. irrespective of the plumage, males just have more of this gene than females, according to your figure. Can you comment on why that might be? I.e. if you compare unmanipulated unornamented males and females, their plumage does not differ, but their CYP2J19 expression does. What do you make of it?

258 – it’s important to note that you only tested this in unornamented males

260 – rephrase “molecular model” to a “mechanistic hypothesis”

264-266 great sentence

268 – change “showing” to “investigating”

274-276 the second part of this sentence seems redundant

281 – I get what you are saying, but this technically is not assessing evolution, but the proximate

basis of color. It can certainly inform evolutionary hypotheses, but those would need to be tested across phylogenies or populations.

283 – again, I don't think we have resolved melanin all that well

286-288 – There are always more genes influencing something, but we have to ask to what degree they are actually explaining variation in the nature. What % of variation is explained by these genes? Little? Most?

290-295 – I think that we can all agree that RNAseq is the new norm and can help, I don't think you need that much text to convince us about it. Maybe just say that we need transcriptome-wide studies to identify the other genes.

295-296 – I am not sure what you mean by “robust framework for focused experimental studies”

296-299 – how exactly does your study help understanding sexual selection? Again, I agree, but I would like to see concrete suggestions. For example, is CYP2J19 involved in other processes in the body?

Figures

Figure 1: The x axis labels on figure 1 is different from the terms you use to describe ornamented vs unornamented males in the text – can you make the terms the same?

Figure 3: Again, it seems that unornamented males have a higher expression of CYP than do females, so I am not sure it is fair to call their levels “low” in this figure

Decision letter (RSPB-2019-2630.R0)

13-Feb-2020

Dear Ms Khalil:

I am writing to inform you that your manuscript RSPB-2019-2630 entitled "Testosterone regulates CYP2J19-linked carotenoid signal expression in male red-backed fairywrens (*Malurus melanocephalus*)" has, in its current form, been rejected for publication in Proceedings B.

This action has been taken on the advice of referees, who have recommended that substantial revisions are necessary. With this in mind we would be happy to consider a resubmission, provided the comments of the referees are fully addressed. However please note that this is not a provisional acceptance.

To upload a resubmitted manuscript, log into <http://mc.manuscriptcentral.com/prsb> and enter your Author Centre, where you will find your manuscript title listed under "Manuscripts with

Decisions." Under "Actions," click on "Create a Resubmission." Please be sure to indicate in your cover letter that it is a resubmission, and supply the previous reference number.

Sincerely,
Dr Locke Rowe
mailto: proceedingsb@royalsociety.org

Associate Editor
Board Member: 1
Comments to Author:

This manuscript describes a study, that combines correlational and experimental data, a powerful combination, to investigate the mechanistic link between circulating testosterone, gene expression and ornamental phenotype. The manuscript received three expert reviews that vary widely in their assessment of the study and the manuscript. Several critical issues were raised: whether age of the experimental males might be a potential confounding variable; the correlational test on circulating carotenoid level variability includes a mixed model where a strong interaction phenotype* age is mentioned – this is not further discussed, but it might be important to interpret the experimental results?; how does re-growth of plucked feathers compare to natural moult, and how do the male categories compare in terms of timing and timeline of feather replacement. Clearly, more details on the sampling and experimental groups are needed and more details on, and/or additional statistical analyses. Also, a better justification of study design is needed (why this time of year, why only liver gene expression); what are the potential consequences of the fact that gene expression of T-implanted birds was only compared to non-manipulated birds, with no induced moult (i.e. the difference in gene expression could not only be due to T, but to a very different physiological state?). Moreover, while parts of the manuscript are very clear, all reviewers highlighted sections where the clarity of the explanations needs to be improved, and the breadth and depth of the discussion and theoretical framing. The latter should also include a frank assessment of study limitations, in terms of design and sample size and, tied in with that, clear future directions. All reviewers made many useful suggestions for revisions.

Reviewer(s)' Comments to Author:

Referee: 1

Comments to the Author(s)

Here, Khalil et al. describe an empirical study that demonstrates the role of testosterone in mediating the expression of the “redness” gene, which in turn influences red ornamental colouration in male fairywrens. The greatest strength of this study is that the results don't settle for correlations between natural levels of variation, but also suggest a direct causal relationship (through experimentally manipulation of testosterone in a subset of birds). The red-backed fairywren is a particularly amenable system for such questions because the testosterone-colouration relationship is perhaps clearer than in other birds with carotenoid-based colouration (where studies, I think, get a bit muddier and contradictory). Overall, I found this to be a clear, convincing, and fascinating manuscript, and my comments are mostly just comments, with a few points for extra clarification.

First paragraph: I found this to be one of the most clear, succinct, and accurate summaries of the carotenoid literature I've read in a recent manuscript. (And actually, the entire introduction is remarkably readable and informative!)

Lines 96-98: When I first learned about this system, I thought that any male in any given year could be either brown or ornamented morph – which, I suppose, is more or less true. But, I remember being surprised to learn that brown males are essentially always younger males and that ornamented males rarely (?) return to the unornamented breeding morph. If any of this is

accurate, I think it's worth clarifying briefly in the text because it does change how we interpret colouration, signalling, and polymorphism in this species.

Lines 166-167: Maybe this is described in results, but how many birds were measured in multiple years?

Line 176: Does it really count as a "moult" if the birds are just growing back that small region of feathers?

Line 188: I imagine there's a story here... what happened to those birds? Or were they just too crafty to recapture?

Line 197: But the testosterone-implanted males would have been growing in an ornamented patch of feathers, right?

Line 202: How does this compare to unimplanted (but ornamented) males?

Lines 225-233: Just to clarify: you didn't take plasma carotenoid measurements from any of the implanted or sacrificed birds, right? So you didn't also test the effect of implantation on carotenoids? Or (and see next comment) are the implanted birds grouped with the ornamented males? [This comes up again on 262-264, where it's not entirely clear if the full pathway of T -> CYP2J19 expression -> ketocarotenoid concentration -> colouration was measured in the implanted, sacrificed birds, or whether different pieces of the puzzle were put together from different birds (which is, of course, quite fine)]

Lines 236-239: The figure clarifies this, but I think it's worth also clarifying in this text where the implanted birds are categorized (as ornamented males?).

Discussion: It could be interesting (and relevant) to mention that Koch, Josefson, Hill 2017 (Biol Rev) propose mito-carotenoid-hormone mechanistic connections, and Hill et al. 2019 (RSPB) provides some empirical evidence for the mito-carotenoid-CYP2J19 connection... Now, your study of course doesn't deal with mitochondria, but it does begin to flesh out the hormone-carotenoid-CYP2J19 connection – which in turn provides some insight into the mito relationships as well!

Referee: 2

Comments to the Author(s)

Comments to editors

In my view, this article addresses an interesting question, the relationship between ketolase gene expression level and red colouration intensity at the within-species level. The article, moreover, proposes to test the testosterone influence on this gene, which is a nice idea. Accordingly, the title suggests that authors have demonstrated that the expression of the ketolase gene involved in yellow-to-red carotenoid conversion is influenced by testosterone levels. However, this assertion is based in a poorly made experiment with very low sample size (only three implanted birds were studied and compared to non-manipulated birds). Any indirect effect (e.g. age, condition) or merely choice can have deeply affected the result. Other methodological problems are also described below.

Major points

1. The model species seems to be quite particular. The authors previously published a study (Lindsay et al 2011 PlosOne) where 2-year-old males were implanted with testosterone during the pre-breeding moult period (August-Sept). Eight T-males produced a redder plumage than seven same-age implanted controls. They argued that this constitutes a particular case where testosterone does not inhibit moult, such as it occurs in other bird species where only a post-

breeding moult takes place. Surprisingly, in contrast to that study, the authors now implanted their birds in the middle of the breeding season (November). Why this?

2. Perhaps to cope with this problem (no active moulting period), they plucked 10 feathers from the back. This would mean that birds would be stimulated to produce red ketocarotenoids for feathers. However, it is not clear if all the birds were able to moult the plucked feathers. Authors only mentioned that the 10-12 d period between implanting date and recapture is a time period that “allowed to re-growth” plucked feathers. Please, clarify.

3. Importantly, since the three sham-operated controls (also plucked) were not recaptured, other non-manipulated birds were used as controls. However, here we cannot discriminate if the supposed increase in CYP2J19 expression among the three T-males was due to high testosterone levels or to the fact of being engaged in producing new feathers. In other words, why should we expect that any of those birds captured only once were activating the ketolase to produce any pigment to feathers if they were not moulting at all?

Moreover, we cannot be sure how the surgery could have influenced the synthesis of redder carotenoids (which are powerful antioxidants).

4. I was also unable to know the age of those few birds used in the implant experiment and subsequent comparisons. The precedent correlational study on 160 birds reported an important role for age. The S1 fig suggests that older birds were able to circulate red ketocarotenoids (see also de legend). Then, the experimental birds should be same-age birds to be compared, reducing this source of variability.

5. Were birds used as controls in the experiment chosen non-randomly? If exactly three birds were used for comparisons, this suggests that authors selected some birds from their larger captures. What was the criterion here? This is important as this could induce some unconscious bias.

6. Recent studies are showing that CYP2J19 can also be expressed at the follicles, not only at the liver. Authors must review Hanlu Twyman’s studies (including his public PhD Thesis; see table 6.6.) where different bird taxa were assessed. This means that CYP2J19 expression at the liver could not be so relevant. Surprisingly, authors had the opportunity of detecting and quantifying the gene in re-growth follicles, but no information is reported.

The site of conversion is also relevant to interpret the results of the correlational study on plasma carotenoid values. Were red ketocarotenoids in plasma the same molecules giving colour to feathers or could they be perhaps subsequently transformed at the follicles?

7. The higher circulating level of red ketocarotenoids in red (ornamented) birds (Fig S1) could be a consequence of phenology. Could the moulting period be established at the individual level (maybe by flight feathers)? In that case, phenotype differences in the days elapsed from capture to the start of the pre-breeding moult should be tested. For instance, dull males could have been sampled farther from their individual moulting time, leading to lower circulating carotenoids. In that case, blood carotenoid levels would not be related to feather colour but to differences in phenology.

8- The birds were captured in two consecutive years in the correlational study on circulating carotenoids. Since the year factor is significant (Table S1), the authors should report sample sizes for each phenotype class each year, providing some chi-q test to know if they were imbalanced. In such a hypothetical case, phenotype-related effects could be, at least partially, due to year variability. If true, year x phenotype should be tested.

9. The significant interaction between phenotype and age in the same model is very relevant as it affects those conclusions only taken from analyzing the phenotype factor. This must be addressed in the main text, with some figure. What is the change in the slope between plasma total red carotenoids and age among the phenotypes? and what it means? Fig S1 is not a suitable representation. Authors must report dispersion plots with slopes for each phenotype type separately, and in the main text (I assume that age was a covariate, i.e. not a 7-level class variable). In the present version, nothing can be concluded about this interaction. I can only suspect that a positive correlation exists between carotenoid levels and age, and the slope is steeper among dull birds, older animals being able to circulate more red carotenoids. That influence of age is

relevant to understand the testosterone experiment (pointed above). Please, note that precedent Lindsay et al 2011 demonstrating the influence of exogenous testosterone on feather colouration was made on second-year birds only.

10. The statistical test performed to interpret the testosterone manipulation engaged 12 individuals and a 4-level factor. Degrees of freedom are however 3,7 (L237). Why this?

11. Moreover, parametric ANOVA requires to meet the normality assumption. Was this the case? With so reduced sample size (particularly in pairwise comparisons: $n = 3$ vs 3) perhaps non-parametric tests should be preferable. In any event, in my view, the sample size is so short that any indirect uncontrolled effect (above) could strongly have affected the results (or merely choice).

An ethical note could here be considered as liver gene expression requires animal killing. However, the accepted rules of bioethics emphasize the fact that sample sizes should be enough large to avoid obtaining inconsistent results requiring to repeating the experiment. Moreover, it is surprising that CYP expression in feather follicles was not explored, which should remove any ethical concern.

12. How feather carotenoid composition was related to plasma composition? are red ketocarotenoids in blood the same that those in feathers? This is important to deduct if other transformations are also made in follicles. In that case, CYP2J19 should also have been assessed there.

13. What are the units in Fig 2? authors mention that they test delta-delta cq , but was is this? the simpler $deltacq$ should be represented. Is the Ct reversed to directly indicate the gene expression level?

Other points

1. L57-59. Please, rewrite. Unclear.

2. L81. Endocrine regulation “underlines” phenotypes.... Please, clarify the meaning.

3. In my view, Peters 2007 BioEssays review on the integration of carotenoid-based signalling and testosterone-controlling effects on sexual signals should be checked and perhaps cited (DOI 10.1002/bies.20563)

4. L142-145. Taking into account the apparent phenotype variability, was the scoring method repeatable? I nonetheless understand that authors removed from the analyses some (11) birds classified in the middle of two classes, which should avoid any doubt.

5. Please, provide a reference to briefly explain how skull ossification was used to know the age of the birds.

6. L156. Analyses containing only known-age birds would have lower stat power (lower sample size; less 23%). Therefore, they can provide quite different results. Please, report them at least as supplementary material.

7. L177. How the peanut oil was hardened? by low temperature?

8. L203 the authors only provide testosterone levels for T-implanted birds at recapture date. Were these birds blood sampled at the first capture? this would allow knowing if T-levels indeed increased at the individual level. Anyway, plasma T-levels of other male birds should also be reported to compare them (though see comments on low sample size and stats).

9. L225-226. In the correlational study on circulating carotenoid levels, the sample of ornamented versus unornamented males was biased to the former 3:1. How was this considered statistically? The authors should test homoscedasticity. Moreover, authors should test normality as carotenoid concentrations are often biased with few animals reporting very high levels. This could require log-transformations. Nothing is commented in this regard.

10. L232. $p < 0.1$ does not mean a significant effect. Is this a mistake?

11. The authors should report the full information about the model used to test differences in circulating carotenoid levels. A table including F and P values for each term in the model with df should be reported, including some stat for the random term (was it significant?).

12. Fig s2. in the X-axis it should be mentioned "T-implant unornamented male"
 13. Fig. 3. If the condition could influence testosterone levels and hence CYP, why body condition (size-corrected body mass) was not tested in the model on 160 birds?
 14. Moreover, why testosterone in plasma was not assessed in these birds?

Referee: 3

Comments to the Author(s)

General comments

This is a very interesting and necessary study that uses experimental approaches to demonstrate the mechanistic link between previously observed effect of T on carotenoid color. The results show that testosterone affects CYP2J19 expression unornamented males, and that, because CYP2J19 is involved in carotenoid synthesis, and plumage color correlates with carotenoid synthesis, that this gene provides a link between T and plumage color. The story would be more complete and stronger, if we could link the level of CYP2J19 expression to the actual plumage score. This is a test I think that the authors can do – by putting the phenotypes into categories, potentially interesting variation is lost. With small sample size, such a correlation may be hard to detect, however, especially since most of the variation in CYP2J19 is explained by sex (see below an additional comment on this).

An additional area where authors could make this study stronger is by being more specific about the implications of this study. The phenotypic integration angle is great, but I think that often in the general literature it is used as a vague umbrella term. I would ask the authors to speculate exactly how defining specific genes that regulate traits may contribute to the understanding of phenotypic integration and the evolution of integrated traits.

Specific comments

Abstract:

Great abstract, very succinct and clear. The only thing I am not sure about what are the “new avenues of investigation” for the link between hormones, signaling, and reproductive success. My understanding is that the link between T and color was shown before – what new avenues are you proposing given that now you have identified one of the candidate genes that may mediate this effect? Studies on the evolution of this gene or its promoters and enhancers? This may be addressed in the discussion, but it would be nice to mention that here, so that the last sentence is not a just-so statement.

Introduction:

50 – I would rephrase “less attention...” to something in the line of “more could be done” – I think that there has been plenty of research on the mechanisms of carotenoid production, e.g. Badyaev’s papers (which hasn’t been cited here and probably should).

62 – I would argue that we still don’t really know how certain melanin types are made (e.g. pheomelanin). You can solve this by saying “some endogenously synthesized pigments such as eumelanin” and then cite a study.

76 – CYP2J19 expression in liver? Or developing feathers?

79 – I agree that studies of wild birds are important, but I would still like you to explicitly say why.

90 – Add that understanding the mechanisms for carotenoid color regulation may provide better insights into how phenotypic integration works. Also, you are painting a very rosy picture here about integration, but here have been some studies that have found that some hormones show poor evidence of integration (e.g. Garamszegi et al 2012 Ethology). Which makes it even MORE important to study the mechanisms.

93-96 – Move citations to the end of the sentence?

99 – maybe rephrase to “driven by the higher rates of extrapair paternity in ornamented males”

109-110 – I think a stronger way to say this would be “...these endocrine processes regulate gene expression”, as steroid receptors are transcription factors.

111 - Insert commas around "known a priori...selection". but that may be a personal preference - it may make the sentence a bit less jammed.

L113 - change "explore" to "test"

113-128 - is there any way you can get data about the effect of T on the actual ketocarotenoid concentration? Or the correlation between T and ketocarotenoids? That would unite the two pieces of information here - you have correlation between ketocarotenoids and plumage, and the effect of T on CYP2J19, but it would be great if you can relate T to ketocarotenoids. If yes, can you cite it here? This is not a make it or break it, but if you have the data it would be great to see them.

Methods

152 - were the intermediate males that you excluded still molting? Not clear here. If they are done molting, why did you exclude them from the analysis?

167 - why "both"? It reads like you are running just one model. What is the difference between the two?

184 - does reference 49 validate these implants in this species? I am not sure how you knew that the implants you gave the birds resulted in the increase of T close to the breeding-season levels. But I appreciate the fact that you are trying to keep the hormone levels in reasonable biological range.

184 - did you use trocar syringe or forceps to insert the implant?

203 - how do these values relate to the natural concentration of T in the breeding season?

214 - did you validate that GADPH expression did not differ between treatments?

215 - is CYP2J19 the only GOI you tested using qPCR?

Results

235 - looking at the figures, it seems that there is a general effect of sex on the CYP levels.

However, that is not something you can explicitly test using your current anova model design, where the males are separated into 3 groups. You have a small sample size, so it may decrease your power, but I wonder what results you'd get if you ran a lm with sex being separate from the plumage score? It would get tricky because you only have T implants in the unornamented males, so the model might have to be nested with respect to that. I understand why you are doing anova here as you do, but each group in your current anova is identified by 3 factors (sex, plumage, and T treatment). The tukey approach allows you the test the T treatment of course, but it doesn't allow an overall sex comparison. But I also understand that the sex comparison here is not the primary objective.

Discussion

General comment: I would like you to discuss the fact that the CYP2J19 levels are higher in the unimplanted unornamented males compared to the females - i.e. irrespective of the plumage, males just have more of this gene than females, according to your figure. Can you comment on why that might be? I.e. if you compare unmanipulated unornamented males and females, their plumage does not differ, but their CYP2J19 expression does. What do you make of it?

258 - it's important to note that you only tested this in unornamented males

260 - rephrase "molecular model" to a "mechanistic hypothesis"

264-266 great sentence

268 - change "showing" to "investigating"

274-276 the second part of this sentence seems redundant

281 - I get what you are saying, but this technically is not assessing evolution, but the proximate basis of color. It can certainly inform evolutionary hypotheses, but those would need to be tested across phylogenies or populations.

283 - again, I don't think we have resolved melanin all that well

286-288 - There are always more genes influencing something, but we have to ask to what degree they are actually explaining variation in the nature. What % of variation is explained by these genes? Little? Most?

290-295 – I think that we can all agree that RNAseq is the new norm and can help, I don't think you need that much text to convince us about it. Maybe just say that we need transcriptome-wide studies to identify the other genes.

295-296 – I am not sure what you mean by “robust framework for focused experimental studies”

296-299 – how exactly does your study help understanding sexual selection? Again, I agree, but I would like to see concrete suggestions. For example, is CYP2J19 involved in other processes in the body?

Figures

Figure 1: The x axis labels on figure 1 is different from the terms you use to describe ornamented vs unornamented males in the text – can you make the terms the same?

Figure 3: Again, it seems that unornamented males have a higher expression of CYP than do females, so I am not sure it is fair to call their levels “low” in this figure

Author's Response to Decision Letter for (RSPB-2019-2630.R0)

See Appendix A.

RSPB-2020-1687.R0

Review form: Reviewer 3

Recommendation

Accept with minor revision (please list in comments)

Scientific importance: Is the manuscript an original and important contribution to its field?

Excellent

General interest: Is the paper of sufficient general interest?

Good

Quality of the paper: Is the overall quality of the paper suitable?

Excellent

Is the length of the paper justified?

Yes

Should the paper be seen by a specialist statistical reviewer?

No

Do you have any concerns about statistical analyses in this paper? If so, please specify them explicitly in your report.

No

It is a condition of publication that authors make their supporting data, code and materials available - either as supplementary material or hosted in an external repository. Please rate, if applicable, the supporting data on the following criteria.

Is it accessible?

N/A

Is it clear?

N/A

Is it adequate?

N/A

Do you have any ethical concerns with this paper?

No

Comments to the Author

General comments. The authors have done a good job of thoroughly addressing the reviewer feedback. The manuscript now addresses the limitations of the study design and sample sizes, and the data analyses has been updated to incorporate reviewer suggestions. I think that the findings are fascinating, but I also agree that the true importance of them will be clearer once we know more about the genetic and hormonal control of carotenoid metabolism, and their incorporation into the feathers in this (and other) species.

There are two issues with the manuscript would like authors to address. First, as I've commented below, in the first round of reviews, I asked about the validation of the GAPDH housekeeping gene, and the authors stated that this gene has been used in many other studies. However, I would still like to see data that show that the absolute expression of GAPDH did not differ between the treatments in these birds, which would validate its use as a housekeeping gene in this study. Second, I think that the first sentence of the discussion invokes a causal relationship that the study cannot address. Please rephrase (see a more specific comment below).

Specific comments

Line 53: The trade-off idea would be clearer if you said, for example, "which suggests a trade-off" instead of "and theory suggests a trade-off".

Line 73: change "linked" to "have so far linked"

Line 74: change to "CYP2J19 to aberrant red coloration in domesticated birds"

Line 76: change "further" to "however" to highlight the contrast between the previous and the following sentence.

Line 84. Start a new sentence with "Androgens.."

Line 92. Citation 44 refers to glucocorticoids, so change "role of androgens" to "role of steroid hormones"

Line 98. You haven't yet postulated this link, so I would change this to "system for linking these two lines of inquiry and asking if testosterone is causally linked to CYP2J19 expression"

Line 178. I assume that CYP2J19 plays a role in the production of all of these ketocarotenoids?

Line 201. Add that the testosterone suspension was then added to the wax mixture and stirred.

Line 202. What was syringe type/ volume?

Line 243. In the first round of reviews, I asked about the validation of the GAPDH housekeeping gene, and the authors stated that this gene has been used in many other studies. However, I would still like to see data that show that the absolute expression of GAPDH did not differ between the treatments, which would validate its use as a housekeeping gene in this study.

Line 283-285. Given the limitations of this study (i.e. not having data on the relationship between CYP2J19 and plumage), you need to tone down this sentence. Specifically, while T does change gene expression causally related to red plumage in other species, you do not have direct experimental evidence that this gene controls coloration this species. So, change “testosterone regulates gene expression to produce sexually selected red plumage in male red-backed fairywrens” to “testosterone regulates gene expression implicated in the production of sexually selected red plumage in male red-backed fairywrens”, or something along the lines.

Line 306. You should add Evans et al. 2000 Beh. Ecol. Sociobiol. citation here which shows that testosterone implants increase bib size in house sparrows.

Line 355. Typo. Is there a word missing in “any effect of by”?

Lines 360-361. Reword this sentence. Long separation between CYP2J19 and “with hue of”, and also I think there is a noun missing after “extend”.

Line 405. Indicate Table S3 here too.

Decision letter (RSPB-2020-1687.R0)

13-Aug-2020

Dear Ms Khalil

I am pleased to inform you that your manuscript RSPB-2020-1687 entitled "Testosterone regulates CYP2J19-linked carotenoid signal expression in male red-backed fairywrens (*Malurus melanocephalus*)" has been accepted for publication in Proceedings B.

The referee(s) have recommended publication, but also suggest some minor revisions to your manuscript. Therefore, I invite you to respond to the referee(s)' comments and revise your manuscript. Because the schedule for publication is very tight, it is a condition of publication that you submit the revised version of your manuscript within 7 days. If you do not think you will be able to meet this date please let us know.

- 1) A text file of the manuscript (doc, txt, rtf or tex), including the references, tables (including captions) and figure captions. Please remove any tracked changes from the text before submission. PDF files are not an accepted format for the "Main Document".

2) A separate electronic file of each figure (tiff, EPS or print-quality PDF preferred). The format should be produced directly from original creation package, or original software format. PowerPoint files are not accepted.

3) Electronic supplementary material: this should be contained in a separate file and where possible, all ESM should be combined into a single file. All supplementary materials accompanying an accepted article will be treated as in their final form. They will be published alongside the paper on the journal website and posted on the online figshare repository. Files on figshare will be made available approximately one week before the accompanying article so that the supplementary material can be attributed a unique DOI.

Sincerely,
Dr Locke Rowe

Associate Editor

Comments to Author:

The authors did a thorough revision and provided detailed responses to all queries. In the course of these revisions they also identified and fixed a few errors in data and statistical analyses that ultimately improved the apparent quality of the experiment. The revisions were seen by one of the previous reviewers, and their remaining comments should be addressed. Regarding the request of the second reviewer to provide information on the absolute expression of GAPDH, I appreciate that this exact information may not be forthcoming, but in that case I suggest the authors give other further and detailed information on the assay validation (e.g. repeatability of GAPDH Ct, CYP Ct, and derived estimates) to reassure the reader that the difference between the unornamented and T-implanted unornamented males could not possibly be due to differences in GAPDH expression. Also, it should be acknowledged that due to the small sample size, the absence of an effect of presence of an implant (yes vs. no) nested within 4 phenotypes is tenuous, beyond what visual inspection of the plot can reveal. The expression 'we confirmed' in l. 248 and l.250 should be toned down. Finally, in l. 343, if the reference added here at the suggestion of the reviewer is indeed out of scope, the authors should feel free to not include it.

Reviewer(s)' Comments to Author:

Referee: 3

Comments to the Author(s).

General comments. The authors have done a good job of thoroughly addressing the reviewer feedback. The manuscript now addresses the limitations of the study design and sample sizes, and the data analyses has been updated to incorporate reviewer suggestions. I think that the findings are fascinating, but I also agree that the true importance of them will be clearer once we know more about the genetic and hormonal control of carotenoid metabolism, and their incorporation into the feathers in this (and other) species.

There are two issues with the manuscript would like authors to address. First, as I've commented below, in the first round of reviews, I asked about the validation of the GAPDH housekeeping gene, and the authors stated that this gene has been used in many other studies. However, I would still like to see data that show that the absolute expression of GAPDH did not differ between the treatments in these birds, which would validate its use as a housekeeping gene in this study. Second, I think that the first sentence of the discussion invokes a causal relationship that the study cannot address. Please rephrase (see a more specific comment below).

Specific comments

Line 53: The trade-off idea would be clearer if you said, for example, "which suggests a trade-off" instead of "and theory suggests a trade-off".

Line 73: change "linked" to "have so far linked"

Line 74: change to "CYP2J19 to aberrant red coloration in domesticated birds"

Line 76: change "further" to "however" to highlight the contrast between the previous and the following sentence.

Line 84. Start a new sentence with "Androgens.."

Line 92. Citation 44 refers to glucocorticoids, so change "role of androgens" to "role of steroid hormones"

Line 98. You haven't yet postulated this link, so I would change this to "system for linking these two lines of inquiry and asking if testosterone is causally linked to CYP2J19 expression"

Line 178. I assume that CYP2J19 plays a role in the production of all of these ketocarotenoids?

Line 201. Add that the testosterone suspension was then added to the wax mixture and stirred.

Line 202. What was syringe type/volume?

Line 243. In the first round of reviews, I asked about the validation of the GAPDH housekeeping gene, and the authors stated that this gene has been used in many other studies. However, I would still like to see data that show that the absolute expression of GAPDH did not differ between the treatments, which would validate its use as a housekeeping gene in this study.

Line 283-285. Given the limitations of this study (i.e. not having data on the relationship between CYP2J19 and plumage), you need to tone down this sentence. Specifically, while T does change gene expression causally related to red plumage in other species, you do not have direct experimental evidence that this gene controls coloration this species. So, change "testosterone regulates gene expression to produce sexually selected red plumage in male red-backed fairywrens" to "testosterone regulates gene expression implicated in the production of sexually selected red plumage in male red-backed fairywrens", or something along the lines.

Line 306. You should add Evans et al. 2000 Beh. Ecol. Sociobiol. citation here which shows that testosterone implants increase bib size in house sparrows.

Line 355. Typo. Is there a word missing in "any effect of by"?

Lines 360-361. Reword this sentence. Long separation between CYP2J19 and "with hue of", and also I think there is a noun missing after "extend".

Line 405. Indicate Table S3 here too.

Author's Response to Decision Letter for (RSPB-2020-1687.R0)

See Appendix B.

Decision letter (RSPB-2020-1687.R1)

19-Aug-2020

Dear Ms Khalil

I am pleased to inform you that your manuscript entitled "Testosterone regulates CYP2J19-linked carotenoid signal expression in male red-backed fairywrens (*Malurus melanocephalus*)" has been accepted for publication in Proceedings B.

Open Access

Paper charges

Sincerely,

Proceedings B

Appendix A

Response to Reviewer Comments:

Testosterone regulates CYP2J19-linked carotenoid signal expression in male red-backed fairywrens (*Malurus melanocephalus*)

Sarah Khalil, Joseph F. Welklin, Kevin J. McGraw, Jordan Boersma, Hubert Schwabl, Michael S. Webster, Jordan Karubian

RSPB-2019-2630

Associate Editor

Board Member: 1

Comments to Author:

Comment: This manuscript describes a study, that combines correlational and experimental data, a powerful combination, to investigate the mechanistic link between circulating testosterone, gene expression and ornamental phenotype. The manuscript received three expert reviews that vary widely in their assessment of the study and the manuscript.

Response: We thank the editor and reviewers for their comments. We have made substantial changes to the manuscript and feel that it is greatly improved. In particular, we feel that we have been able to effectively respond to the key concerns of Reviewer 2 related to our experimental design and interpretation.

Comment: Several critical issues were raised: whether age of the experimental males might be a potential confounding variable; the correlational test on circulating carotenoid level variability includes a mixed model where a strong interaction phenotype* age is mentioned – this is not further discussed, but it might be important to interpret the experimental results?;

Response: We plotted the model for our circulating carotenoid data and found that there is only an effect of age within unornamented males, but no effect in either females or ornamented males (Figure 1). We go into more detail about this in our response to Reviewer 2's major comment #9. We have also added discussion about the age*phenotype interaction in unornamented males for the circulating carotenoids in lines 332-341.

Comment: how does re-growth of plucked feathers compare to natural moult, and how do the male categories compare in terms of timing and timeline of feather replacement.

Response: There is good reason to believe that the mechanisms (e.g. genes expressed) for the regrowth of plucked feathers is similar to natural moult; we discuss this more detail in response to reviewer comments below. Individuals used in our testosterone experiment (both controls and testosterone-implanted individuals) molted in feathers around the same time. At time of collection, all individuals that were previously plucked were in similar stages of feather regrowth, and we added this point into the manuscript in lines 211-213 (methods).

Reviewer 2 had suggested using flight feathers as a way to infer when individuals molt, but because fairywrens only molt their flight feathers during their post-breeding season

molt, we cannot use this method. We discuss this in more detail in our response to Reviewer 2 (major comment #7).

Comment: Clearly, more details on the sampling and experimental groups are needed and more details on, and/or additional statistical analyses. Also, a better justification of study design is needed (why this time of year, why only liver gene expression);

Response: We agree, and have added more details throughout the manuscript in response to all comments from reviewers, with specific line numbers for each change provided in our detailed responses below. **Please note these line numbers refer to the clean manuscript, not the track-changes document.** In addition, we explain in more detail the justification for both the time of year of our study and our focus on liver gene expression in lines 192-195 and lines 135-136 of the revised manuscript.

Comment: what are the potential consequences of the fact that gene expression of T-implanted birds was only compared to non-manipulated birds, with no induced moult (i.e. the difference in gene expression could not only be due to T, but to a very different physiological state?).

Response: In responding to this comment, we discovered that 2 of the 3 individuals we classified as “unmanipulated dull males” actually had been implanted with a sham control and had feathers plucked at the time of sham implantation; only the third bird was unmanipulated. To account for this change in the underlying data, we ran a new linear model where CYP2J19 expression (log fold change) was the response variable, and whether the individual had an implant (yes vs. no) was nested within phenotype as the predictor variable. There was no interaction effect between phenotype and implant, and including the implant (yes/no) term as a predictor variable did not improve the model fit, suggesting that the presence or absence of having the sham-implant within dull males did not influence the expression data. For the sake of thoroughness, we also ran a separate ANOVA on the model described above and found the same qualitative result: no effect of implant on CYP2J19 expression. We also evaluated raw data and confirmed that the log fold change for the unmanipulated male (2.20) was similar to that of the two sham implant males (1.95 and 2.19), which can be seen in **supplementary figure S2**.

For these reasons, we have retained the same ANOVA between expression and phenotype used in the initial submission (i.e., without implant type yes/no as a predictor term), but have updated the methods to correctly describe the underlying data types and the rationale describe above for not including implant as a term in the model. (see lines 214-216, 248-253 of the revised manuscript, lines 20-34 in the supplementary materials).

We apologize for this mistake and thank the editor and reviewers for leading us to uncover it. This revision also helps to address related concerns about the impact of feather plucking vs. T-implants on the observed patterns. This information increases our confidence that observed differences in CYPJ219 expression are due to the testosterone implant itself, not feather plucking, since the two sham-control males had their feathers plucked (described in line 209-213).

Comment: Moreover, while parts of the manuscript are very clear, all reviewers highlighted sections where the clarity of the explanations needs to be improved, and the breadth and depth of the discussion and theoretical framing. The latter should also include a frank assessment of study limitations, in terms of design and sample size and, tied in with that, clear future directions. All reviewers made many useful suggestions for revisions.

Response: We have made major edits to the discussion, increasing the depth as well as being more clear about the theoretical framework and future directions (please see more detailed responses to Reviewer comments below). We also discuss the limitations of our study, including our smaller sample size for the hormone manipulations in lines 354-357 of the discussion

1) Responses to Referee 1

Comment: First paragraph: I found this to be one of the most clear, succinct, and accurate summaries of the carotenoid literature I've read in a recent manuscript. (And actually, the entire introduction is remarkably readable and informative!)

Response: Thank you for this comment! We very much appreciated the kind words!

Comment: Lines 96-98: When I first learned about this system, I thought that any male in any given year could be either brown or ornamented morph—which, I suppose, is more or less true. But, I remember being surprised to learn that brown males are essentially always younger males and that ornamented males rarely (?) return to the unornamented breeding morph. If any of this is accurate, I think it's worth clarifying briefly in the text because it does change how we interpret colouration, signalling, and polymorphism in this species.

Response: Yes those statements about brown males are accurate, as is reflected in our circulating carotenoid data where we only have unornamented males that are one or two years old. We have added more information about how plumage phenotype is associated with breeding status, age and condition in lines 102-110.

Comment: Lines 166-167: Maybe this is described in results, but how many birds were measured in multiple years?

Response: Thirteen individuals were sampled in both years, and we have added this information to line 184.

Comment: Line 176: Does it really count as a "moult" if the birds are just growing back that small region of feathers?

Response: Thank you for pointing this out - we have changed this to "feather replacement" instead of moult (now line 197).

Comment: Line 188: I imagine there's a story here... what happened to those birds? Or were they just too crafty to recapture?

Response: As described in our response to the editor above, we actually were only unable to catch one bird. We discovered that 2 of the 3 individuals we classified as “unmanipulated dull males” actually had been implanted with a sham control and had feathers plucked at the time of sham implantation, and we have addressed the consequences for interpretation of this separately, in our comments to the editor. The third implanted bird was unfortunately too crafty to recapture. We attempted to catch that individual at least 4 times, but he had become extremely net-shy (more than we have ever seen in others) and we were just unable to catch him again.

Comment: Line 197: But the testosterone-implanted males would have been growing in an ornamented patch of feathers, right?

Response: Yes, they were growing in red pin feathers and we have added a sentence about that in the methods (lines 211-212).

Comment: Line 202: How does this compare to unimplanted (but ornamented) males?

Response: Testosterone concentrations were similar between the testosterone-implanted and ornamented males, and we have added that information to the manuscript (lines 226-230).

Comment: Lines 225-233: Just to clarify: you didn’t take plasma carotenoid measurements from any of the implanted or sacrificed birds, right? So you didn’t also test the effect of implantation on carotenoids? Or (and see next comment) are the implanted birds grouped with the ornamented males? [This comes up again on 262-264, where it’s not entirely clear if the full pathway of T -> CYP2J19 expression -> ketocarotenoid concentration -> colouration was measured in the implanted, sacrificed birds, or whether different pieces of the puzzle were put together from different birds (which is, of course, quite fine)]

Response: You are correct that we did not take carotenoid measurements from the implanted and sacrificed birds. The full pathway is not measured in the sacrificed birds, but rather based on two separate data sets: the circulating carotenoid data from unmanipulated birds, and the experimental testosterone-implant gene expression data. We clarify this in lines 298 – 301 of the discussion. Being able to directly compare circulating testosterone levels and carotenoid levels is something we ideally would love to do. However, the small size of these birds and associated limitations in the volume of blood we can collect from them makes this challenging. Due to their very small size (6-7 grams), we are only able to collect enough plasma from them to either run an HPLC analysis for carotenoids, or for the testosterone assay, but unable to do both for the same sample.

We were able to collect more blood from the sacrificed individuals – unfortunately though, many plasma samples were either lost or destroyed during transport. We were able to recover some samples for testosterone assays (reported in the manuscript lines 226-230), but we were unable to recover any for HPLC analysis.

Comment: Lines 236-239: The figure clarifies this, but I think it’s worth also clarifying in this text where the implanted birds are categorized (as ornamented males?).

Response: The implanted males are categorized as their own “phenotype” in order to be able to directly compare their CYP2J19 expression to both naturally ornamented males and unimplanted unornamented males. We label these birds as “implanted unornamented males” which we describe in the manuscript on lines 220-223.

Comment: Discussion: It could be interesting (and relevant) to mention that Koch, Josefson, Hill 2017 (Biol Rev) propose mito-carotenoid-hormone mechanistic connections, and Hill et al. 2019 (RSPB) provides some empirical evidence for the mito-carotenoid-CYP2J19 connection... Now, your study of course doesn't deal with mitochondria, but it does begin to flesh out the hormone-carotenoid-CYP2J19 connection—which in turn provides some insight into the mito relationships as well!

Response: We do think mitochondria-carotenoid work by Hill et al. is really interesting and relevant work to the carotenoid field, but outside of the scope of our study. Their focus there was understanding how mitochondria might mediate variation within red signals surrounding carotenoid-signal honesty. We are not testing the honesty the red-plumage in this study, and we do not look at variation within the redness of plumage, but instead are focused on characterizing the mechanisms that produce carotenoid-based plumage more generally (i.e. the shift from brown to red plumage). However, we have now cited the paper in the discussion, on line 343.

2) Responses to Referee 2

Comment: In my view, this article addresses an interesting question, the relationship between ketolase gene expression level and red colouration intensity at the within-species level. The article, moreover, proposes to test the testosterone influence on this gene, which is a nice idea. Accordingly, the title suggests that authors have demonstrated that the expression of the ketolase gene involved in yellow-to-red carotenoid conversion is influenced by testosterone levels. However, this assertion is based in a poorly made experiment with very low sample size (only three implanted birds were studied and compared to non-manipulated birds). Any indirect effect (e.g. age, condition) or merely choice can have deeply affected the result. Other methodological problems are also described below.

Response: Thank you for your helpful comments. We deal with each point on a case by case basis below. We would have liked to use a larger number of birds in this experimental design if we were able to so. Because the design calls for sacrificing individuals, we were significantly constrained in the number of birds we could get approval for to use by the relevant Australian governmental agencies. The study design presented here represents the maximum number of animals we could implant and sacrifice. We also feel that we were able to address the valid concerns and questions raised about age and the effects of plucking vs. testosterone implantation, which has greatly improved the manuscript.

Major points

Comment: 1. The model species seems to be quite particular. The authors previously published a study (Lindsay et al 2011 PlosOne) where 2-year-old males were implanted with testosterone during the pre-breeding moult period (August-Sept). Eight T-males produced a redder plumage than seven same-age implanted controls. They argued that this constitutes a particular case where testosterone does not inhibit moult, such as it occurs in other bird species where only a post-breeding moult takes place. Surprisingly, in contrast to that study, the authors now implanted their birds in the middle of the breeding season (November). Why this?

Response: Thank you for the opportunity to clarify this point- the distinction between pre- and post-nuptial moult is key. Studies in other systems that report that testosterone inhibits moult refer to a delay of onset of a postnuptial molt in seasonally breeding birds of temperate zones. In contrast, testosterone actively induces onset of pre-nuptial moult in fairywrens.

We implanted males during the breeding season to disentangle breeding status from the genetic and hormonal mechanisms that were our primary focus in this study. Specifically brown unornamented males either pair with a female and become a “breeder”, or they remain on the natal territory as an auxiliary “helper”, which in turn has implications for circulating testosterone levels (Lindsay et al. 2009). Moreover, helper males can transition to become a breeding male if they find an available female at any point during the breeding season, which could disrupt the experiment. At this time during the breeding season, males are finished with moult (that is why we plucked feather to assess the coloration of regrown feathers with and without testosterone implants). To control for potential effects of breeding status (i.e., breeder vs. helper) as well as to avoid any possible change in breeding status during the experiment, we focused on known breeders, which by definition requires conducting the experiment during the breeding season. We have added information explaining this to the methods (lines 192-195).

Comment: 2. Perhaps to cope with this problem (no active moulting period), they plucked 10 feathers from the back. This would mean that birds would be stimulated to produce red ketocarotenoids for feathers. However, it is not clear if all the bird were able to moult the plucked feathers. Authors only mentioned that the 10-12 d period between implanting date and recapture is a time period that “allowed to re-growth” plucked feathers. Please, clarify.

Response: We have clarified that birds that were plucked did exhibit feather regrowth at the time of the second capture (i.e., pin feathers were starting to grow in) on lines 211-213.

Comment: 3. Importantly, since the three sham-operated controls (also plucked) were not recaptured, other non-manipulated birds were used as controls. However, here we cannot discriminate if the supposed increase in CYP2J19 expression among the three T-males was due to high testosterone levels or to the fact of being engaged in producing new feathers. In other words, why should we expect that any of those birds captured only once were activating the ketolase to produce any pigment to feathers if they were not moulting at all?

Response: In responding to this comment, we discovered that 2 of the 3 individuals we classified as “unmanipulated dull males” actually had been implanted with a sham control and had feathers plucked at the time of sham implantation; only the third bird was unmanipulated. To account for this change in the underlying data, we ran a new linear model where CYP2J19 expression (log fold change) was the response variable, and whether the individual had an implant (yes vs. no) was nested within phenotype as the predictor variable. There was no interaction effect between phenotype and implant, and including the implant (yes/no) term as a predictor variable did not improve the model, suggesting that the presence or absence of having the sham-implant within dull males did not influence the expression data. For the sake of thoroughness, we also ran a separate ANOVA on the model described above and found the same qualitative result: no effect of implant on CYP2J19 expression. We also evaluated raw data and confirmed that the log fold change for the unmanipulated male (2.20) was similar to that of the two sham implant males (1.95 and 2.19), which can be seen in **supplementary figure S2**.

For these reasons, we have retained the same ANOVA between expression and phenotype used in the initial submission (i.e., without implant type yes/no as a predictor term), but have updated the methods to correctly describe the underlying data types and the rationale describe above for not including implant as a term in the model. (see lines 214-216, 248-253 of the revised manuscript, lines 20-34 in the supplementary materials).

We apologize for this mistake and thank you for leading us to uncover it. This revision also helps to address related concerns about the impact of feather plucking vs. T-implants on the observed patterns. This information increases our confidence that observed differences in CYPJ219 expression are due to the testosterone implant itself, not feather plucking, since the two sham-control males had their feathers plucked (described in line 209-213).

Comment: Moreover, we cannot be sure how the surgery could have influenced the synthesis of redder carotenoids (which are powerful antioxidants).

Response: As described in the above comment, two of the three control birds did in fact have surgery, indicating that the surgery itself is unlikely to be influencing differences in CYP2J19 expression we observed between control and testosterone implanted males. We apologize again for the error on our part in the original submission and confusion that has caused. In addition, there is increasing evidence that carotenoids may not serve as antioxidants as much or as strongly as previously has been thought.

Comment: 4. I was also unable to know the age of those few birds used in the implant experiment and subsequent comparisons. The precedent correlational study on 160 birds reported an important role for age. The S1 fig suggests that older birds were able to circulate red ketocarotenoids (see also de legend). Then, the experimental birds should be same-age birds to be compared, reducing this source of variability.

Response: We agree with the reviewer that knowing the age of the experimental birds would be an advantage. However, as is often the case with field-based experiments on wild animals, we lack age information for these birds beyond the fact that they were at

least one year old (based on skull ossification). Because we knew we would be sacrificing these birds, we were unable to use individuals that form part of our long-term field study due to impacts on ongoing population monitoring there. We chose birds that were just off of that site – a site we do not consistently monitor and band at, and therefore we unfortunately do not have exact age information for them.

Comment: 5. Were birds used as controls in the experiment chosen non-randomly? If exactly three birds were used for comparisons, this suggests that authors selected some birds from their larger captures. What was the criterion here? This is important as this could induce some unconscious bias.

Response: As noted above, there was only a single bird (not 3) used as an unmanipulated control. Selection was constrained by permitting and use of birds off our main long-term study site. There was no pre-meditated selection of birds – for the no-sham unornamented male, we caught the first unornamented male that we confirmed was paired with a female.

Comment: 6. Recent studies are showing that CYP2J19 can also be expressed at the follicles, not only at the liver. Authors must review Hanlu Twyman’s studies (including his public PhD Thesis; see table 6.6.) where different bird taxa were assessed. This means that CYP2J19 expression at the liver could not be so relevant. Surprisingly, authors had the opportunity of detecting and quantifying the gene in re-growth follicles, but no information is reported.

Response: Thank you for this suggestion – we were unaware of Twyman’s PhD thesis document. We prioritized gene expression in the liver rather than feathers because the high concentrations of ketocarotenoids in the plasma suggest that red-backed fairywrens may be “central” ketoconverters, with the liver as the main site of carotenoid ketolation. We agree that assaying feathers as a potential ketolation site in this species represents an important avenue for future work, and have added this information into our Discussion (line 357-361).

Comment: The site of conversion is also relevant to interpret the results of the correlational study on plasma carotenoid values. Were red ketocarotenoids in plasma the same molecules giving colour to feathers or could they be perhaps subsequently transformed at the follicles?

Response: We agree that looking at feather gene expression makes sense in light of the previous comment, and we have added text in the Discussion that flags this as a priority for future research (line 357-361).

However, we do know that the red feathers of red-backed fairywren are composed of the same four ketocarotenoids we find in the plasma – these results can be found in (Rowe and McGraw 2008), Table 2. We have added a statement about this in the manuscript (line 258-260).

Comment: 7. The higher circulating level of red ketocarotenoids in red (ornamented) birds (Fig S1) could be a consequence of phenology. Could the moulting period be established at the individual level (maybe by flight feathers)? In that case, phenotype

differences in the days elapsed from capture to the start of the pre-breeding moult should be tested. For instance, dull males could have been sampled farther from their individual moulting time, leading to lower circulating carotenoids. In that case, blood carotenoid levels would not be related to feather colour but to differences in phenology.

Response: In this species, when individuals molt into their breeding plumage (the pre-alternate molt for those who molt from brown to brown plumage), they do not molt their flight feathers. Fairywrens only molt their flight feathers once a year during their pre-basic molt after the breeding season. So we unfortunately cannot use flight feathers to establish the molt we are studying for these birds.

Comment: 8- The birds were captured in two consecutive years in the correlational study on circulating carotenoids. Since the year factor is significant (Table S1), the authors should report sample sizes for each phenotype class each year, providing some chi-q test to know if they were imbalanced. In such a hypothetical case, phenotype-related effects could be, at least partially, due to year variability. If true, year x phenotype should be tested.

Response: When rerunning the models to control for heteroscedasticity (brought up in “Other points” #9), we discovered that Year actually did not improve the fit of the model (did not improve AIC by more than 2 and p-value of the Year variable was greater than 0.05) and therefore dropped it from the model. This has been added to the methods (line 185-187). The results of the new model (without Year) are qualitatively the same as the old model (with Year), but we now report the statistics associated with the new model in the results.

Comment: 9. The significant interaction between phenotype and age in the same model is very relevant as it affects those conclusions only taken from analyzing the phenotype factor. This must be addressed in the main text, with some figure. What is the change in the slope between plasma total red carotenoids and age among the phenotypes? and what it means? Fig S1 is not a suitable representation. Authors must report dispersion plots with slopes for each phenotype type separately, and in the main text (I assume that age was a covariate, i.e. not a 7-level class variable. In the present version, nothing can be concluded about this interaction. I can only suspect that a positive correlation exists between carotenoid levels and age, and the slope is steeper among dull birds, older animals being abler to circulate more red carotenoids. That influence of age is relevant to understand the testosterone experiment (pointed above). Please, note that precedent Lindsay et al 2011 demonstrating the influence of exogenous testosterone on feather colouration was made on second-year birds only.

Response: Thank you for pointing this out – we should have added a graph representing our model in the original manuscript, and have added it now (see new Figure 1A). We plotted the ketocarotenoid concentration by age, with the model trend lines through the data. You are correct that, in our model, age is a continuous variable, and we have added that to the methods section on line 181. We have added to our results that there is an interaction between age and ketocarotenoid, and specifically there is an age effect only for unornamented males, where two year-olds have higher concentrations than one year-olds (line 264-268). But as can be seen in this graph,

there is no relationship between age and ketocarotenoid concentration in ornamented males or females. We also discuss our interpretations of these results in lines 332-341.

Comment:10. The statistical test performed to interpret the testosterone manipulation engaged 12 individuals and a 4-level factor. Degrees of freedom are however 3,7 (L237). Why this?

Response: Thank you for catching this mistake. The degrees of freedom should be 3,8 and we have corrected that in the results (line 271).

Comment: 11. Moreover, parametric ANOVA requires to meet the normality assumption. Was this the case? With so reduced sample size (particularly in pairwise comparisons: $n = 3$ vs 3) perhaps non-parametric tests should be preferable.

Response: For qPCR data, normality assumptions are met. The dependent variable is normally distributed in each group, and we tested this using a Shapiro-Wilk test in R using the `shapiro.test` function. We also tested for constant variance of residuals (homoscedasticity) using a Breusch Pagan test, and found constant variance. We have added this information to the methods (line 250).

Comment: In any event, in my view, the sample size is so short that any indirect uncontrolled effect (above) could strongly have affected the results (or merely choice). An ethical note could here be considered as liver gene expression requires animal killing. However, the accepted rules of bioethics emphasize the fact that sample sizes should be enough large to avoid obtaining inconsistent results requiring to repeating the experiment.

Response: As noted above, we were limited in our sample collection size due to collecting restrictions set by the Australia Environment Protection Authority who issue our collecting permits. This represents the maximum allowance we could receive. We also note that a sample size of 3 per treatment has been used in several comparable studies on wild and captive birds, including (Lopes et al. 2016) highly cited study in Current Biology. Some studies even published with less than 3 samples per treatment; for example in (Twyman et al. 2018) some species had only 1 or 2 samples tested, and in (Xu et al. 2016) there were only 2 samples per treatment.

In our study, within each phenotype, CYP2J19 log fold change tend to cluster tightly together (as can be seen by the small standard errors in each phenotype in Figure 2), which suggests little variance and a low probability of major uncontrolled factors of influence.

Comment: Moreover, it is surprising that CYP expression in feather follicles was not explored, which should remove any ethical concern.

Response: As above (comment 6), we acknowledge that this represents a valuable direction for future research and have noted this in line 357-361.

Comment: 12. How feather carotenoid composition was related to plasma composition? are red ketocarotenoids in blood the same that those in feathers? This is important to deduct if other transformations are also made in follicles. In that case, CYP2J19 should also have been assessed there.

Response: As described in our response to comment 6 above, we do know that the red feathers of red-backed fairywren are composed of the same four ketocarotenoids we find in the plasma – these results can be found in (Rowe and McGraw 2008), Table 2. We have added a statement about this in the manuscript (line 258-260). We plan to address these related points of gene expression in the feather in the near future.

Comment: 13. What are the units in Fig 2? authors mention that they test delta-delta cq, but was is this? the simpler deltacq should be represented. Is the Ct reversed to directly indicate the gene expression level?

Response: There are no units when using the delta-delta ct method since it is a measure of relative expression (specifically log fold change here, as written on the y-axis). We feel that it is better to represent the result this way (instead of just the delta ct) because we can compare results to females (we set the average female log fold change to 0 to better visualize the difference in gene expression between all four phenotypes). We have added details about this in the supplementary materials methods (lines 8-18 of the supplement) to more clearly explain that.

Other points

Comment: 1. L57-59. Please, rewrite. Unclear.

Response: We have removed this sentence.

Comment: 2. L81. Endocrine regulation “underlines” phenotypes.... Please, clarify the meaning.

Response: Here, “underlies” is used similarly as “influences”, and we give examples as to how endocrine regulation underlies phenotypes in the next sentence. Any wording suggestions from the reviewer are welcome (now line 83).

Comment: 3. In my view, Peters 2007 BioEssays review on the integration of carotenoid-based signalling and testosterone-controlling effects on sexual signals should be checked and perhaps cited (DOI 10.1002/bies.20563)

Response: Thank you for pointing this out. We are familiar with this paper, and mistakenly did not cite it in our original submission. We have added it to the revised manuscript (line 95).

Comment: 4. L142-145. Taking into account the apparent phenotype variability, was the scoring method repeatable? I nonetheless understand that authors removed from the analyses some (11) birds classified in the middle of two classes, which should avoid any doubt.

Response: All members of the field team practiced scoring plumage and were tested for several days with photos, as well as in the field, to ensure repeatability of plumage scoring. Given that we are “binning” the birds into one of two clearly defined, discrete categories and we removed ambiguous/intermediate individuals from the analysis (as noted by the reviewer), we feel confident about this scoring method.

Comment: 5. Please, provide a reference to briefly explain how skull ossification was used to know the age of the birds.

Response: We have added a reference to the methods (line 170).

Comment: 6. L156. Analyses containing only known-age birds would have lower statistical power (lower sample size; less 23%). Therefore, they can provide quite different results. Please, report them at least as supplementary material.

Response: Results from the analysis with only known-age birds are qualitatively the same as the analysis with the full data set. We noted this in the text of the original submission (now lines 171-172), but have also added these results to the supplementary material (Table S1) to avoid any confusion.

Comment: 7. L177. How the peanut oil was hardened? by low temperature?

Response: Yes, the peanut oil was frozen. We have added this to the methods (line 198).

Comment: 8. L203 the authors only provide testosterone levels for T-implanted birds at recapture date. Were these birds blood sampled at the first capture? this would allow knowing if T-levels indeed increased at the individual level. Anyway, plasma T-levels of other male birds should also be reported to compare them (though see comments on low sample size and stats).

Response: We unfortunately do not have testosterone levels at the time of implantation. Because these birds are so small, we were unable to safely collect blood samples of sufficient volume to measure testosterone repeatedly over the course of the implant period.

Comment: 9. L225-226. In the correlational study on circulating carotenoid levels, the sample of ornamented versus unornamented males was biased to the former 3:1. How was this considered statistically? The authors should test homoscedasticity. Moreover, authors should test normality as carotenoid concentrations are often biased with few animals reporting very high levels. This could require log-transformations. Nothing is commented in this regard.

Response: Thank you for bringing this up. We originally did not consider homoscedasticity. After running our new model (described in “Major Points” #8 above), we visually inspected the residuals in our model for heteroscedasticity. We then used the varIdent function in R to control for heterogeneity of variance between groups. As described in (Zuur, Ieno, and Elphick 2010), it is more appropriate to visually inspect residuals for homogeneity then do an actual test for normality since most tests are not actually appropriate for testing normality of residuals. We have added this information to the methods (line XX). The results of the model remained qualitatively the same, but we now report statistics associated with the new model in the results.

Comment: 10. L232. $p < 0.1$ does not mean a significant effect. Is this a mistake?

Response: Thank you for catching this. This was a mistake and we have corrected it in the results (line 184-185).

Comment: 11. The authors should report the full information about the model used to test differences in circulating carotenoid levels. A table including F and P values for each term in the model with df should be reported, including some stat for the random term (was it significant?).

Response: We have added the full summary of the model to the supplementary materials (Table S2).

Comment: 12. Fig s2. in the X-axis it should be mentioned "T-implant unornamented male"

Response: We have changed this.

Comment: 13. Fig. 3. If the condition could influence testosterone levels and hence CYP, why body condition (size-corrected body mass) was not tested in the model on 160 birds?

Response: Because we don't actually have testosterone data for the 160 birds we tested in the circulating carotenoid model, it seemed outside the scope of the model to also test body condition. We are more interested in the difference between phenotypes, but will further investigate individual differences (including condition) in the future.

Comment: 14. Moreover, why testosterone in plasma was not assessed in these birds?

Response: We did measure plasma testosterone levels in both the testosterone-implanted birds (originally in the manuscript), and the ornamented birds (added into the manuscript lines 229-230), but unfortunately do not have meaningful testosterone data for the control unornamented males or the females because we were unable to get those samples while collecting, or they were lost during transport (added this note to lines 230-232).

3) Responses to Referee 3

Comment: The story would be more complete and stronger, if we could link the level of CYP2J19 expression to the actual plumage score. This is a test I think that the authors can do – by putting the phenotypes into categories, potentially interesting variation is lost. With small sample size, such a correlation may be hard to detect, however, especially since most of the variation in CYP2J19 is explained by sex (see below an additional comment on this).

Response: Ideally we would love to be able to link individual CYP2J19 expression to individual feather color (hue), but it is not possible in this study because our sample sizes per group are so small. We only have three individuals (the ornamented males) for which we could measure feather color. The individuals that we implanted with testosterone did start growing in red feathers, but they were just erupting from the feather sheath at time of collection and therefore are not large enough to measure hue using a spectrophotometer. However, we have flagged this (investigating finer-grained

relationships with hue of red plumage) as a topic for future work in our discussion (lines 357-261).

Comment: An additional area where authors could make this study stronger is by being more specific about the implications of this study. The phenotypic integration angle is great, but I think that often in the general literature it is used as a vague umbrella term. I would ask the authors to speculate exactly how defining specific genes that regulate traits may contribute to the understanding of phenotypic integration and the evolution of integrated traits.

Response: We thank the reviewer for suggesting specific areas of the manuscript where we can be more specific about the implications of the study. We have made changes throughout the manuscript (more details in the specific comment below), and we believe these changes have greatly improved our resubmission.

Specific comments

Abstract:

Comment: Great abstract, very succinct and clear. The only thing I am not sure about what are the “new avenues of investigation” for the link between hormones, signaling, and reproductive success. My understanding is that the link between T and color was shown before – what new avenues are you proposing given that now you have identified one of the candidate genes that may mediate this effect? Studies on the evolution of this gene or its promoters and enhancers? This may be addressed in the discussion, but it would be nice to mention that here, so that the last sentence is not a just-so statement.

Response: Thank you for this comment, and we have decided to delete that last sentence, and instead focus on the more concrete results of the study. The abstract now ends: “This is the first time that hormonal regulation of a specific genetic locus has been linked to carotenoid production in a natural context, revealing how endocrine mechanisms produce sexual signals that shape reproductive success” (lines 32-35).

Introduction:

Comment: 50 – I would rephrase “less attention...” to something in the line of “more could be done” – I think that there has been plenty of research on the mechanisms of carotenoid production, e.g. Badyaev’s papers (which hasn’t been cited here and probably should).

Response: We have changed this to “However, many aspects of the underlying mechanisms of carotenoid colour production remain unclear”, and we have added relevant Badyaev citations (lines 49-51).

Comment: 62 – I would argue that we still don’t really know how certain melanin types are made (e.g. pheomelanin). You can solve this by saying “some endogenously synthesized pigments such as eumelanin” and then cite a study.

Response: Done (line 63).

Comment: 76 – CYP2J19 expression in liver? Or developing feathers?

Response: In the liver - we have added this to the sentence (line 78).

Comment: 79 – I agree that studies of wild birds are important, but I would still like you to explicitly say why.

Response: We have added a statement to the last sentence that more explicitly explains why we need to study this specifically in wild birds (line 79-82).

Comment: 90 – Add that understanding the mechanisms for carotenoid color regulation may provide better insights into how phenotypic integration works. Also, you are painting a very rosy picture here about integration, but here have been some studies that have found that some hormones show poor evidence of integration (e.g. Garamszegi et al 2012 Ethology). Which makes it even MORE important to study the mechanisms.

Response: We have added these to the end of this paragraph, including the Garamszegi citation and another citation showing limited evidence T on integration (Lipshutz et al. 2019) (lines 91-96).

Comment: 93-96 – Move citations to the end of the sentence?

Response: Done (line 102).

Comment: 99 – maybe rephrase to “driven by the higher rates of extrapair paternity in ornamented males.”

Response: We have rephrased this as suggested (line 111).

Comment: 109-110 – I think a stronger way to say this would be “...these endocrine processes regulate gene expression”, as steroid receptors are transcription factors.

Response: We agree, and have rephrased as suggested (line 122).

Comment: 111 – Insert commas around “known a priori....selection”. but that may be a personal preference – it may make the sentence a bit less jammed.

Response: We have removed that part of the sentence.

Comment: L113 – change “explore” to “test”

Response: Done (line 125).

Comment: 113-128 – is there any way you can get data about the effect of T on the actual ketocarotenoid concentration? Or the correlation between T and ketocarotenoids? That would unite the two pieces of information here – you have correlation between ketocarotenoids and plumage, and the effect of T on CYP2J19, but it would be great if you can relate T to ketocarotenoids. If yes, can you cite it here? This is not a make it or break it, but if you have the data it would be great to see them.

Response: Being able to directly compare circulating testosterone levels and carotenoid levels is something we ideally would love to do. However, the major reason we have not been able to is because of the size of these birds, and in turn the limitations we have in extracting blood from them. Due to their very small size (6-7 grams), we are only able to obtain enough plasma per sampling to either run an HPLC analysis for carotenoids, or run the testosterone assay, but not both.

Methods

Comment: 152 – were the intermediate males that you excluded still molting? Not clear here. If they are done molting, why did you exclude them from the analysis?

Response: We excluded these males because we were uncertain as to whether or not they had completed molt, and because we felt that the inclusion of a small number of intermediate males distinct from the fully unornamented vs. ornamented types used in the experiment detracted from, rather than added to, resolving the core goals of the study.

Comment: 167 – why “both”? It reads like you are running just one model. What is the difference between the two?

Response: Thank you for catching this - this was a mistake in our manuscript. There is only one model and we changed the manuscript to reflect that (line 188).

Comment: 184 – does reference 49 validate these implants in this species? I am not sure how you knew that the implants you gave the birds resulted in the increase of T close to the breeding-season levels. But I appreciate the fact that you are trying to keep the hormone levels in reasonable biological range.

Response: Reference 49 is not a validation of the actual implant, but a reference to what the normal range of testosterone is in the breeding season. We apologize for the confusion. Validation studies have not been published, but we report the plasma testosterone concentrations for testosterone-implanted individuals to demonstrate that they fall within the natural range of testosterone for ornamented males (line 205).

Comment: 184 – did you use trocar syringe or forceps to insert the implant?

Response: We used forceps, and have added this information to the methods (line 205).

Comment: 203 – how do these values relate to the natural concentration of T in the breeding season?

Response: They are within the natural range of testosterone for ornamented males in the breeding season. We have added that statement here with a reference (Lindsay et al. 2009) (line 229), as well as testosterone values for the ornamented males in this study (lines 229-230).

Comment: 214 – did you validate that GAPDH expression did not differ between treatments?

Response: We were unable to validate that GAPDH expression did not differ between treatments. However, GAPDH is commonly used (if not the most commonly used gene) as a reliable housekeeping gene across different types of treatments in birds (Gazda et al. 2020; Lopes et al. 2016; Toomey et al. 2017, 2018). We therefore believed it would be the best for our qPCR analysis, and we have added these citations to the supplementary methods (lines 11-12 of supplemental).

Comment: 215 – is CYP2J19 the only GOI you tested using qPCR?

Response: Yes CYP2J19 was the only gene of interest we tested with qPCR.

Results

Comment: 235 – looking at the figures, it seems that there is a general effect of sex on the CYP levels. However, that is not something you can explicitly test using your current anova model design, where the males are separated into 3 groups. You have a small sample size, so it may decrease your power, but I wonder what results you'd get if you ran a lm with sex being separate from the plumage score? It would get tricky because you only have T implants in the unornamented males, so the model might have to be nested with respect to that. I understand why you are doing anova here as you do, but each group in your current anova is identified by 3 factors (sex, plumage, and T treatment). The tukey approach allows you the test the T treatment of course, but it doesn't allow an overall sex comparison. But I also understand that the sex comparison here is not the primary objective.

Response: Thank you for this suggestion. Yes, since a sex comparison was not the primary objective of the study, we did not initially test for it. We ran a lm on our expression data to try to answer your question: $\text{lm}(\log\text{foldchange} \sim \text{sex} + \text{sex}/\text{treatment})$, where sex was either male or female, and treatment was either yes or no, where only males had the option for yes (the unornamented t-treated males.) These were the results of the anova on the lm:

	Df	Sum Sq	Mean Sq	F value	Pr(>F)
sex	1	14.7120	14.7120	148.3946	6.774e-07 ***
sex:treatment	1	0.3954	0.3954	3.9882	0.07692
Residuals	9	0.8923	0.0991		

Therefore we do see an overall sex effect, but not a significant sex*treatment interaction effect, which we believe is because the treatment birds had expression levels in between the control unornamented males and the ornamented males, therefore hiding the treatment effect when you are just looking at all males together in comparison to females.

Again, because the sex comparison was not the main objective, but we have added this analysis to the supplemental (lines 36-47 of supplemental, and table S3).

Discussion

Comment: General comment: I would like you to discuss the fact that the CYP2J19 levels are higher in the unimplanted unornamented males compared to the females – i.e. irrespective of the plumage, males just have more of this gene than females, according to your figure. Can you comment on why that might be? I.e. if you compare unmanipulated unornamented males and females, their plumage does not differ, but their CYP2J19 expression does. What do you make of it?

Response: We have added a detailed paragraph discussing this in the discussion (321-341). In short, unornamented males paired with females have higher levels of testosterone than females or auxiliary (non-breeding helper) males, and we believe that increased testosterone levels lead to the higher CYP2J19 expression we see in our unornamented males in this study.

Comment: 258 – it’s important to note that you only tested this in unornamented males

Response: We have added that point to this sentence (line 292).

Comment: 260 – rephrase “molecular model” to a “mechanistic hypothesis”

Response: Done, thank you for the suggestion (line 299).

Comment: 264-266 great sentence

Response: Thank you, we appreciate the comment!

Comment: 268 – change “showing” to “investigating”

Response: Done, thank you for the suggestion (line 305).

Comment: 274-276 the second part of this sentence seems redundant

Response: We agree and have streamlined this sentence (line 309-311).

Comment: 281 – I get what you are saying, but this technically is not assessing evolution, but the proximate basis of color. It can certainly inform evolutionary hypotheses, but those would need to be tested across phylogenies or populations.

Response: Here we are not really referring to macro-evolutionary processes, but rather how evolution has shaped the stable polymorphism in male breeding phenotypes in red-backed fairywrens, because it has persisted. We have added a sentence after this to be more explicit about how these findings can be also be important for other systems with strong sexual selection (line 317-320).

Comment: 283 – again, I don’t think we have resolved melanin all that well

Response: We have removed our statement about melanin.

Comment: 286-288 – There are always more genes influencing something, but we have to ask to what degree they are actually explaining variation in the nature. What % of variation is explained by these genes? Little? Most?

Response: This is true, and we added “to varying degrees” at the end of this sentence (line 348) to incorporate that idea here.

Comment: 290-295 – I think that we can all agree that RNAseq is the new norm and can help, I don’t think you need that much text to convince us about it. Maybe just say that we need transcriptome-wide studies to identify the other genes.

Response: We have cut this section down (lines 352-354).

Comment: 295-296 – I am not sure what you mean by “robust framework for focused experimental studies”

Response: We have removed this sentence.

Comment: 296-299 – how exactly does your study help understanding sexual selection? Again, I agree, but I would like to see concrete suggestions. For example, is CYP2J19 involved in other processes in the body?

Response: We have changed this sentence so it's not about sexual selection per se, but how sexually selected traits are regulated, which is the focus of this study (line 361-365).

Figures

Comment: Figure 1: The x axis labels on figure 1 is different from the terms you use to describe ornamented vs unornamented males in the text – can you make the terms the same?

Response: Yes, we have changed the labels to be the same in all figures – ornamented male, unornamented male, and female.

Comment: Figure 3: Again, it seems that unornamented males have a higher expression of CYP than do females, so I am not sure it is fair to call their levels “low” in this figure

Response: We have changed the figure to “lower” and “higher” CYP2J19 expression, instead of just low and high.

Works cited:

- Gazda, Małgorzata Anna et al. 2020. “A Genetic Mechanism for Sexual Dichromatism in Birds.” *Science* 12(6496):1–6.
- Lindsay, Willow R., Michael S. Webster, Claire W. Varian, and Hubert Schwabl. 2009. “Plumage Colour Acquisition and Behaviour Are Associated with Androgens in a Phenotypically Plastic Tropical Bird.” *Animal Behaviour* 77(6):1525–32. Retrieved (<http://dx.doi.org/10.1016/j.anbehav.2009.02.027>).
- Lipshutz, S. E., E. M. George, A. B. Bentz, and K. A. Rosvall. 2019. “Evaluating Testosterone as a Phenotypic Integrator: From Tissues to Individuals to Species.” *Molecular and Cellular Endocrinology* 496(July):110531. Retrieved (<https://doi.org/10.1016/j.mce.2019.110531>).
- Lopes, Ricardo J. et al. 2016. “Genetic Basis for Red Coloration in Birds.” *Current Biology* 26(11):1427–34.
- Rowe, Melissa and Kevin J. McGraw. 2008. “Carotenoids in the Seminal Fluid of Wild Birds : Interspecific Variation in Fairy-Wrens.” *The Condor* 110(4):694–700.
- Toews, David P. L., Natalie R. Hofmeister, and Scott A. Taylor. 2017. “The Evolution and Genetics of Carotenoid Processing in Animals.” *Trends in Genetics* 33(3):171–82. Retrieved (<http://dx.doi.org/10.1016/j.tig.2017.01.002>).
- Toomey, Matthew B. et al. 2017. “High-Density Lipoprotein Receptor SCARB1 Is Required for Carotenoid Coloration in Birds.” *Proceedings of the National Academy of Sciences* 114(20):5219–24. Retrieved (<http://www.pnas.org/lookup/doi/10.1073/pnas.1700751114>).
- Toomey, Matthew B. et al. 2018. “A Non-Coding Region near Follistatin Controls Head Color Polymorphism in Gouldian Finch.” *Proceedings of the Royal Society B: Biological Sciences*.
- Twyman, Hanlu, Maria Prager, Nicholas I. Mundy, and Staffan Andersson. 2018. “Expression of a Carotenoid-Modifying Gene and Evolution of Red Coloration in Weaverbirds (Ploceidae).” *Molecular Ecology* 27(2):449–58. Retrieved (<http://doi.wiley.com/10.1111/mec.14451>).

- Xu, Xiaoqin et al. 2016. "Transcriptomic Analysis of Different Stages of Pigeon Ovaries by RNA-Sequencing." *Molecular Reproduction and Development* 83(7):640–48.
- Zuur, Alain F., Elena N. Ieno, and Chris S. Elphick. 2010. "A Protocol for Data Exploration to Avoid Common Statistical Problems." *Methods in Ecology and Evolution* 1(1):3–14.

Appendix B

Response to Referees:

Testosterone regulates CYP2J19-linked carotenoid signal expression in male red-backed fairywrens (*Malurus melanocephalus*)

Sarah Khalil, Joseph F. Welklin, Kevin J. McGraw, Jordan Boersma, Hubert Schwabl, Michael S. Webster, Jordan Karubian

RSPB-2020-1687

We thank the editor and the referee for their feedback. The track-changes version of the final manuscript can be found in this document after the response to comments. Line numbers in the comments refer to line numbers in the final clean manuscript.

Associate Editor

Comments to Author:

The authors did a thorough revision and provided detailed responses to all queries. In the course of these revisions they also identified and fixed a few errors in data and statistical analyses that ultimately improved the apparent quality of the experiment. The revisions were seen by one of the previous reviewers, and their remaining comments should be addressed. Regarding the request of the second reviewer to provide information on the absolute expression of GAPDH, I appreciate that this exact information may not be forthcoming, but in that case I suggest the authors give other further and detailed information on the assay validation (e.g. repeatability of GAPDH Ct, CYP Ct, and derived estimates) to reassure the reader that the difference between the unornamented and T-implanted unornamented males could not possibly be due to differences in GAPDH expression.

Response: Thank you for this feedback. As you suspected, we do not have information for absolute expression of GAPDH, but we have run several tests to ensure that GAPDH expression does not differ by treatment.

First, we ran a linear mixed effect model that included GAPDH Ct as the response variable and phenotype (female vs. unornamented male vs. testosterone implanted unornamented male vs. ornamented male) as the predictor variable. We added individual sample as a random effect to control for repeated measures since each sample was run in triplicate during qPCR (except for that female sample in duplicate; please see note below). We tested the model for significance with an ANOVA using the aov function in R, and found no significant effect of phenotype on GAPDH Ct. The effect of the fixed predictor variable (phenotype) on GAPDH Ct using this model, with sample as a random effect, can now be found in Table S3.

Second, we calculated the intra-assay coefficient of variance (CV) for the GAPDH, and CYP2J19 qPCR assays and found they were 1.13% and 2.13% respectively. Generally, intra-assay %CV should be below 10 to assume sufficient repeatability of the assay. The calculations for this will be added as another sheet in our qPCR data deposited in Dryad.

Third, we graphed the Ct values for GAPDH and CYP2J19 across phenotypes as a way to visualize derived estimates/ranges, which can be found in supplementary figure S2 (and this raw Ct data is deposited in Dryad). As can be seen, GAPDH values

visually do not differ between phenotypes (which was confirmed statistically with the above described model). Therefore, differences in log fold change must be driven by actual differences in CYP2J19 expression. We have added all of this to the supplementary materials (“Testing for differences in GAPDH Ct between phenotypes” section, lines 9-20)

Note: When running these tests, we realized that one of the female samples was only run in duplicate for CYP2J19 qPCR assay, instead of triplicate like the rest of the samples. All values in the manuscript are correct and reflect this, but we had neglected to include this information in the earlier versions. We have corrected this omission in the current version (lines 243-244, methods).

Also, it should be acknowledged that due to the small sample size, the absence of an effect of presence of an implant (yes vs. no) nested within 4 phenotypes is tenuous, beyond what visual inspection of the plot can reveal. The expression ‘we confirmed’ in l. 248 and l.250 should be toned down.

We have added this acknowledgment in the supplementary methods (lines 45-47), and changed the line with “we confirmed” in the manuscript to “We found no effect of presence or absence of the sham implant on gene expression within unornamented males (see electronic supplementary material, Methods), and no evidence of homoscedasticity (Breusch Pagan test, $p>0.05$).” (lines 249-251).

Finally, in l. 343, if the reference added here at the suggestion of the reviewer is indeed out of scope, the authors should feel free to not include it.

The reference does link CYP2J19 to red plumage color, so we are fine to include it as there is no requirement to also discuss the mitochondrial aspect of that paper.

Reviewer(s)' Comments to Author:

Referee: 3

Comments to the Author(s).

General comments. The authors have done a good job of thoroughly addressing the reviewer feedback. The manuscript now addresses the limitations of the study design and sample sizes, and the data analyses has been updated to incorporate reviewer suggestions. I think that the findings are fascinating, but I also agree that the true importance of them will be clearer once we know more about the genetic and hormonal control of carotenoid metabolism, and their incorporation into the feathers in this (and other) species.

There are two issues with the manuscript would like authors to address. First, as I've commented below, in the first round of reviews, I asked about the validation of the GAPDH housekeeping gene, and the authors stated that this gene has been used in many other studies. However, I would still like to see data that show that the absolute expression of GAPDH did not differ between the treatments in these birds, which would validate its use as a housekeeping gene in this study. Second, I think that the first

sentence of the discussion invokes a causal relationship that the study cannot address. Please rephrase (see a more specific comment below).

Response:

We thank the reviewer for all their clear and specific suggestions below and helping improve the writing of this manuscript. We have addressed all the comments below and changed them as suggested.

Specific comments

Comment: Line 53: The trade-off idea would be clearer if you said, for example, “which suggests a trade-off” instead of “and theory suggests a trade-off”.

Response: Done.

Line 73: change “linked” to “have so far linked”

Response: Done.

Line 74: change to “CYP2J19 to aberrant red coloration in domesticated birds”

Response: Done.

Line 76: change “further” to “however” to highlight the contrast between the previous and the following sentence.

Response: Done.

Line 84. Start a new sentence with “Androgens..”

Response: Done.

Line 92. Citation 44 refers to glucocorticoids, so change “role of androgens” to “role of steroid hormones”

Response: Done.

Line 98. You haven’t yet postulated this link, so I would change this to “system for linking these two lines of inquiry and asking if testosterone is causally linked to CYP2J19 expression”

Response: Because we thought having the word “link” twice here was a bit repetitive, we changed this sentence to “The red-backed fairywren (*Malurus melanocephalus*) provides a useful study system for linking these two lines of inquiry and assessing how CYP2J19 and testosterone regulation may interact to control intraspecific variation in the expression of a carotenoid-based signal.” (lines 97-99)

Line 178. I assume that CYP2J19 plays a role in the production of all of these ketocarotenoids?

Response: Yes, the presumption is that since CYP2J19 is a ketolase, it plays a role in the production of keto-carotenoids. Putative pathways of dietary carotenoid ketolation into ketocarotenoids can be found in (McGraw, Hill, Stradi, & Parker, 2001).

Line 201. Add that the testosterone suspension was then added to the wax mixture and stirred.

Response: Done.

Line 202. What was syringe type/volume?

The co-author who made the testosterone implants could not find the information for the exact type of syringe used, but stated that the top was cut off so the diameter of the tip was 2mm. We have added that information to the methods in line 203.

Line 243. In the first round of reviews, I asked about the validation of the GAPDH housekeeping gene, and the authors stated that this gene has been used in many other studies. However, I would still like to see data that show that the absolute expression of GAPDH did not differ between the treatments, which would validate its use as a housekeeping gene in this study.

We explain in detail how we tested that GAPDH expression does not differ between treatments in response to the editor's comments above:

We have run several tests to ensure that GAPDH expression does not differ by treatment.

First, we ran a linear mixed effect model that included GAPDH, Ct as the response variable and phenotype (female vs. unornamented male vs. testosterone implanted unornamented male vs. ornamented male) as the predictor variable. We added individual sample as a random effect to control for repeated measures since each sample was run in triplicate during qPCR (except for that female sample in duplicate; please see note above). We tested the model for significance with an ANOVA using the aov function in R, and found no significant effect of phenotype on GAPDH, Ct. The effect of the fixed predictor variable (phenotype) on GAPDH, Ct using this model, with sample as a random effect, can now be found in Table S3.

Second, we calculated the intra-assay coefficient of variance (CV) for the GAPDH, and CYP2J19 qPCR assays and found they were 1.13% and 2.13% respectively. Generally, intra-assay %CV should be below 10 to assume sufficient repeatability of the assay. The calculations for this will be added as another sheet in our qPCR data deposited in Dryad.

Third, we graphed the Ct values for GAPDH and CYP2J19 across phenotypes as a way to visualize derived estimates/ranges, which can be found in supplementary figure S2 (and this raw Ct data is deposited in Dryad). As can be seen, GAPDH values visually do not differ between phenotypes (which was confirmed statistically with the above described model). Therefore, differences in log fold change must be driven by actual differences in CYP2J19 expression. We have added all of this to the supplementary materials.

Line 283-285. Given the limitations of this study (i.e. not having data on the relationship between CYP2J19 and plumage), you need to tone down this sentence. Specifically, while T does change gene expression causally related to red plumage in other species, you do not have direct experimental evidence that this gene controls coloration this species. So, change "testosterone regulates gene expression to produce sexually selected red plumage in male red-backed fairywrens" to "testosterone regulates gene

expression implicated in the production of sexually selected red plumage in male red-backed fairywrens”, or something along the lines.

Response: Done.

Line 306. You should add Evans et al. 2000 Beh. Ecol. Sociobiol. citation here which shows that testosterone implants increase bib size in house sparrows.

Response: Done.

Line 355. Typo. Is there a word missing in “any effect of by”?

We have edited this sentence to “We note the small sample size for our gene expression experiment (n=3 per phenotype, due to permitting restrictions), and we attempted to minimize any effect of this by only using breeding individuals to reduce phenology differences as well as collecting all birds in a relatively small time period (described in methods).” (line 355-358).

Lines 360-361. Reword this sentence. Long separation between CYP2J19 and “with hue of”, and also I think there is a noun missing after “extend”.

We changed the second part of the sentence to “and investigate finer-grained relationships between CYP2J19 expression and hue of red plumage to extend beyond the presence/absence-of-coloration approach we employ in the current study.” (lines 361-363).

Line 405. Indicate Table S3 here too.

Response: Done, and we have added the other supplementary table and figure captions.

References:

McGraw, K. J., Hill, G. E., Stradi, R., & Parker, R. S. (2001). The influence of carotenoid acquisition and utilization on the maintenance of species-typical plumage pigmentation in male American goldfinches (*Carduelis tristis*) and northern cardinals (*Cardinalis cardinalis*). *Physiological and Biochemical Zoology*, 74(6), 843–852. <https://doi.org/10.1086/323797>